# ICONGETM v1.0 – Flexible NUOPC-driven two-way coupling via ESMF exchange grids between the unstructured-grid atmosphere model ICON and the structured-grid coastal ocean model GETM

Tobias Peter Bauer[1,2], Peter Holtermann[2], Bernd Heinold[1], Hagen Radtke[2], Oswald Knoth[1], and Knut Klingbeil[2]

[1]Leibniz Institute for Tropospheric Research (TROPOS), Permoserstraße 15, 04318 Leipzig
[2]Leibniz Institute for Baltic Sea Research Warnemünde (IOW), Seestraße 15, 18119 Rostock

**Correspondence:** Tobias Peter Bauer (tobias.bauer@tropos.de)

**Abstract.** Two-way model coupling is important for representing the mutual interactions and feedbacks between atmosphere and ocean dynamics. This work presents the development of the two-way coupled model system ICONGETM, consisting of the atmosphere model ICON and the ocean model GETM. ICONGETM is built on the latest NUOPC coupling software with flexible data exchange and conservative interpolation via ESMF exchange grids. With ICON providing a state-of-the-art kernel for numerical weather prediction on an unstructured mesh and GETM being an established coastal ocean model, ICONGETM is especially suited for high-resolution studies. For demonstration purposes the newly developed model system has been applied to a coastal upwelling scenario in the Central Baltic Sea.

## 1 Introduction

In numerous studies, the added value of two-way coupled atmosphere-ocean models has been demonstrated. Interactive model coupling is important for representing the mutual interactions and feedbacks between atmosphere and ocean dynamics (e.g., Chelton and Xie, 2010). The sea surface temperature (SST) of the ocean determines moisture fluxes into the atmosphere and the stability of the atmospheric boundary layer (Fallmann et al., 2019). The modulated surface wind in turn affects surface currents and mixing in the ocean, both altering SST patterns. This air-sea interaction is very dynamic and strongly sensitive to fronts and eddies (Small et al., 2008; Shao et al., 2019). In the coastal ocean, fronts are further pronounced due to upwelling and river run-off. Therefore, especially high-resolution coastal applications, where sharp gradients and small-scale eddies are resolved, can benefit from two-way coupled atmosphere-ocean models.

The atmosphere model COAMPS (Hodur, 1997) and the regional ocean model ROMS (Shchepetkin and McWilliams, 2005) were coupled with the Model Coupling Toolkit (MCT; Larson et al., 2005) for investigating an upwelling event with a 1 km high resolution (Perlin et al., 2007), see Appendix A for list of model acronyms. In the following decade, numerous high resolution studies were performed with the two-way coupled model system COAMPS-NCOM, in which COAMPS was originally coupled via MCT with the coastal ocean model NCOM (Barron et al., 2006). Pullen et al. (2006, 2007) demonstrated the improved skill of the two-way coupled model system during Bora events in the Adriatic Sea, simulated down to a resolution of

4 km in the atmosphere and 2 km in the ocean. With the same resolution and a coupling time step of 12 min, the model system has been applied to the Ligurian Sea and confirmed the importance of the interactive model coupling in the coastal zone (Small et al., 2011). The impact of coastal orography was investigated in a 2 km simulation of Madeira Island (Pullen et al., 2017). Another two-way coupled model system widely applied in high-resolution studies is COAWST (Warner et al., 2010). It couples the atmosphere model WRF (Skamarock et al., 2005), the ocean model ROMS and the wave model SWAN (Booij et al., 1999) with MCT. COAWST has been applied for a realistic hindcast of a storm event over the Gulf of Lion and the Balearic Seas with a resolution of 3 km in the atmosphere and 1.8 km in the ocean (Renault et al., 2012). In another application, a Bora event and the dense water formation in the Adriatic sea with 7 km resolution in the atmosphere and 1 km in the ocean was simulated (Carniel et al., 2016). Both studies investigated the effects of different coupling strategies and demonstrated the benefit of the fully coupled model system. Recently, the high-resolution regional coupled environmental prediction system UKC for the northwest European Shelf has been developed (Lewis et al., 2018, 2019a). On a 1.5 km high resolution, the atmosphere model MetUM (Cullen, 1993; Brown et al., 2012) was coupled with the ocean model NEMO (Madec et al., 2017) via OASIS3-MCT (Valcke et al., 2012; Craig et al., 2017). First results demonstrate reduced bias in SST fields (Lewis et al., 2019b) and impacts on cloud and fog formation over the North Sea (Fallmann et al., 2017, 2019).

Key technical aspects of coupled model systems are the coordinated execution of the individual model components and the data exchange among them. Required infrastructure for time management, communication between different nodes and interpolation between different grids is provided by various coupling software, e.g. MCT, OASIS-MCT, ESMF. Coupling frameworks, like the Earth System Modeling Framework (ESMF; Hill et al., 2004), provide an additional superstructure layer which offers a standardized execution of models as model components and data exchange in coupler components. On top of ESMF, the National Unified Operational Prediction Capability (NUOPC) layer (Theurich et al., 2016) defines generic components which offer a unified and automated driving of coupled model systems. The generic components require only minimum specialization for the individual models, e.g. registration of routines for initialization and time step advance, definition of required import and possible export data. NUOPC automatically negotiates the data exchange between individual model components based on standard names and synonyms from a dictionary. All required information about model grids and distribution among processors is received during runtime from the models. Therefore, models once equipped with a NUOPC-compliant interface can be plugged into any other coupled model system driven by NUOPC, without the need to adapt coupling specifications.

NUOPC supports a seamless data exchange and interpolation between models operating on different grids via so called connectors. In addition, NUOPC offers mediator components to perform e.g. merging, time-averaging and interface flux calculations on a hub between several models. With ESMF/NUOPC, it is also possible to perform these calculations on automatically generated exchange grids. They have been introduced in Balaji et al. (2006) as the union of the vertices of the individual model grids. ESMF provides this functionality for unstructured grids, with the final exchange grid obtained by a triangulation of the union. This triangulation is the basis for conservative interpolation. Moreover, the ESMF exchange grid considers the masking of the original grids, e.g. land/sea masks, and excludes fractions that are not required for the interpolation.

There is an ongoing effort to implement the new NUOPC layer into model systems and equip many popular models with a NUOPC interface under the umbrella of the Earth System Prediction Suite (Theurich et al. (2016); https://www.icams-portal.

gov/resources/espc/esps/table.htm). However, until now, there exists only a limited number of publications about its integration. The functioning of the NUOPC layer within the Regional Earth System Model (RegESM) was described by Turuncoglu (2019). Sun et al. (2019) developed the regional integrated prediction system SKRIPS based on NUOPC, coupling the atmosphere model WRF and the nonhydrostatic ocean model MITgcm (Marshall et al., 1997). Only very recently, a coupled unstructured-grid model application consisting of the ocean model ADCIRC (Luettich, Jr. et al., 1992) and the wave model WAVEWATCH III (WW3DG, 2019) within the NUOPC-based NOAA Environmental Modeling System (NEMS; https://www.emc.ncep.noaa.gov/emc/pages/infrastructure/nems.php) was reported by Moghimi et al. (2020).

Despite the potential of the ESMF exchange grid, its implementation and usage in a mediator component has not been published, yet. This paper presents the newly developed model system ICONGETM, consisting of the atmosphere model ICON (Zängl et al., 2015) and the coastal ocean model GETM (Burchard and Bolding, 2002). With ICON providing a state-of-the-art kernel for numerical weather prediction on an unstructured mesh and GETM being one of the leading Baltic Sea models, a so far missing coupled model system for high-resolution studies in the Baltic has been developed. The model system is based on a NUOPC-Mediator, taking care of the data exchange via an ESMF exchange grid.

First, the technical structure of ICONGETM including a short overview of ICON and GETM as well as the automated coupling with ESMF/NUOPC is described in Sec. 2. The details of the data exchange and interpolation using the ESMF exchange grid are explained in Sec. 3. In Sec. 4, a demonstration of the coupled model system for the Central Baltic Sea is presented. The added value and potential of using the ESMF exchange grid in ICONGETM are discussed in detail in Sec. 5. And finally, the paper is summarized in Sec. 6.

## 2 The coupled model system ICONGETM

### 2.1 The atmospheric model ICON

The ICOsahedral Non-hydrostatic modelling framework (ICON) was developed by the German Weather Service (DWD) and the Max Planck Institute for Meteorology (MPI-M) as a unified modelling system for global numerical weather prediction (NWP) and climate modelling, including exact local mass conservation, mass-consistent tracer transport, a flexible grid nesting capability and the usage of nonhydrostatic Euler equations on global domains (e.g. Gassmann and Herzog, 2008; Dipankar et al., 2015; Zängl et al., 2015; Heinze et al., 2017; Giorgetta et al., 2018; Crueger et al., 2018; Borchert et al., 2018). The details of the model are given in Zängl et al. (2015). They have been summarized in Ullrich et al. (2017) for the dynamical core model inter-comparison project (DCMIP) 2016.

The atmospheric component of ICON allows various user-configurations for different modelling scenarios, e.g. large eddy simulations, numerical weather prediction or climate simulations, by coupling a common dynamical core with different physics packages. The model used in this study is a configuration led by DWD, mainly used for high-resolution NWP applications. Some physics schemes largely inherit the fast-physics parametrizations from the atmosphere model COSMO, see Zängl et al. (2015).

ICON solves the 2-D vector-invariant equations on an icosahedral triangular grid with Arakawa C-grid staggering and terrain-

following vertical discretization. A predictor–corrector scheme is employed, which is explicit in all terms except for those describing the vertical propagation of sound waves. The nesting capability in ICON includes a bisection of the simulation time step from one nest to the other.

The DWD applies ICON as a member of the operative weather forecast system in Germany (DWD, 2019). High-resolutions simulations were conducted to understand the physical feedbacks due to clouds (e.g. Dipankar et al., 2015; Heinze et al., 2017). MPI-M uses the ICON Earth system model (ICON-ESM; e.g. Hanke et al., 2016; Giorgetta et al., 2018; Crueger et al., 2018), where individual model components for the atmosphere (ICON-A), ocean (ICON-O) and land (ICON-L) are coupled with the YAC library (Hanke et al., 2016).

For the coupling in ICONGETM, an interface to ESMF was implemented for the nonhydrostatic NWP core.

## 2.2 The ocean model GETM

The General Estuarine Transport Model (GETM) is an open-source ocean model for coastal and regional applications (www.getm.eu). Originally developed for solving the primitive equations as well as transport equations for temperature and salinity on C-staggered finite volumes (Burchard and Bolding, 2002), it nowadays also offers a non-hydrostatic extension of the dynamic kernel for high-resolution applications (Klingbeil and Burchard, 2013). GETM supports boundary-following vertical coordinates with adaptive interior model layers (Hofmeister et al., 2010; Gräwe et al., 2015). The nonlinear free surface is computed by a split-explicit mode-splitting technique with drying-and-flooding capability, see the review about numerics of coastal ocean models by Klingbeil et al. (2018). GETM uses efficient 2nd-order transport schemes with minimized spurious mixing (Klingbeil et al., 2014). State-of-the-art turbulence closure is provided from the General Ocean Turbulence Model (GOTM; www.gotm.net). Via an interface to the Framework for Aquatic Biogeochemical Models (FABM; https://fabm.net), GETM can act as a hydrodynamic host model for a variety of biogeochemical models. An efficient decomposition into subdomains offers high-performance computing on massively parallel systems for high-resolution and climate-scale simulations (e.g. Gräwe et al., 2019; Lange et al., 2020). For coupling to other models, GETM already provides an interface to ESMF (Lemmen et al., 2018).

## 2.3 Coupling with ESMF/NUOPC

ICONGETM is built on ESMF/NUOPC. It is hierarchically structured into main program, driver, model and coupler components, see Fig. 1. The NUOPC layer controls the execution and interaction of the components by triggering different phases for their Initialization, Run and Finalization. Generic actions are performed automatically and only individual specification routines need to be implemented for the components. The implementation of the NUOPC layer in ICONGETM was inspired by the prototype codes `AtmOcnMedPetListProto`, `AtmOcnTransferGridProto`, `CustomFieldDictionaryProto` and `AtmOcnFDSynoProto` as well as `AtmOcnConProto` from https://earthsystemmodeling.org/nuopc/#prototype-applications.

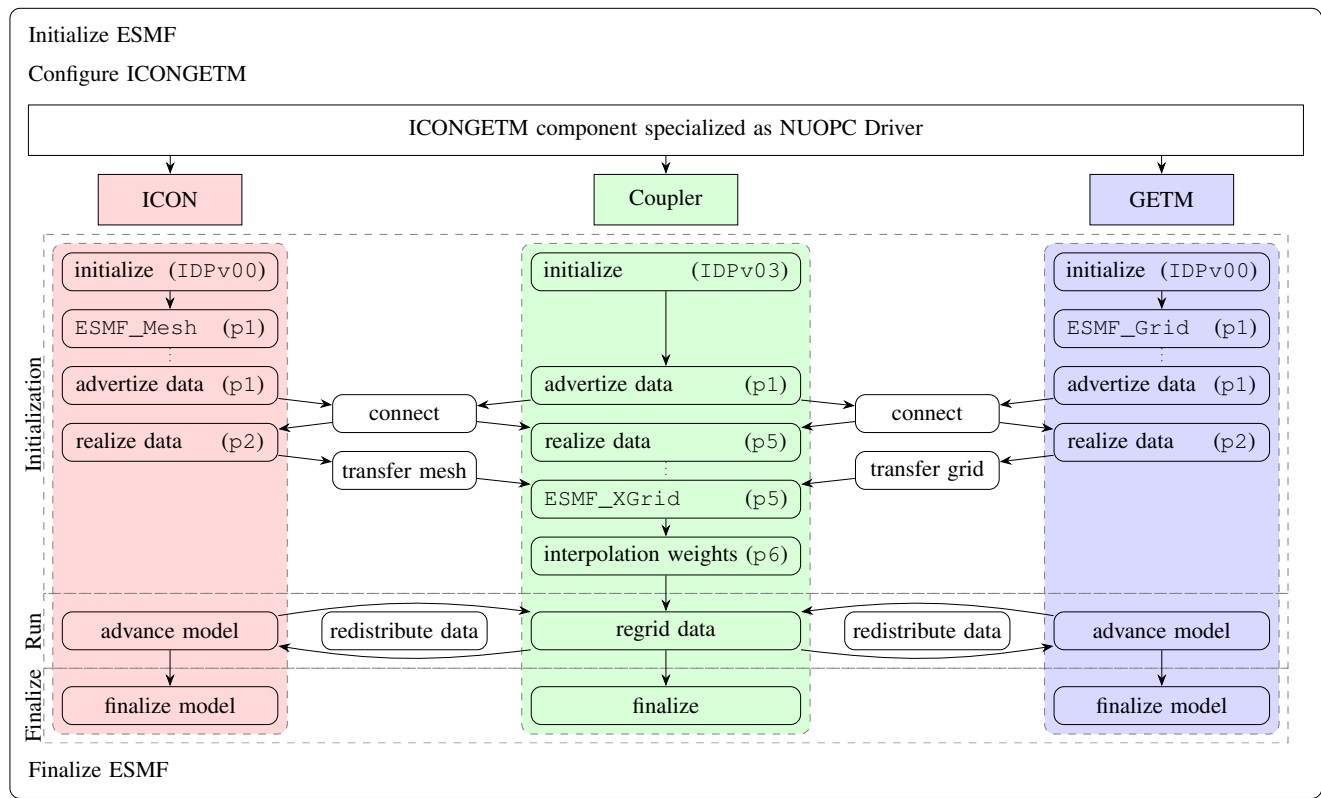

**Figure 1.** Structure of ICONGETM. The ICONGETM component created by the main program is specialized as NUOPC-Driver and consists of NUOPC-Model components for ICON and GETM as well as a NUOPC-Mediator for the Coupler. For all components the implemented specialized routines for initialization, run and finalization are indicated. The initialization phases of the NUOPC-layer are given in parenthesis. Automated generic NUOPC operations are represented by arrows. In the pdf version of this article the central NUOPC components presented in the Figure are linked to the corresponding locations in the source code.

### 2.3.1 Initialization

ICONGETM is initialized and configured in different stages. At first, ESMF itself is initialized. Next, the coupled model is configured from a user-provided configuration file with the number of processes for each model component, the names of the data to be received by each model component as well as the coupling time step. A NUOPC-Driver is applied, which creates

NUOPC-Model components for ICON and GETM as well as a NUOPC-Mediator, which serves as a data exchange component between the model components. The current implementation only supports a concurrent distribution of the components among all available computing units. For the time management, a run sequence defines in which order the mediator and model components will interact during the simulation.

Next, the initialization routines of each NUOPC-Model component are called. They have access to the initializing routines
of the individual models themselves. Additionally, the horizontal grid structures are translated into an ESMF_Grid and

| Quantity | ICON | | Coupler | GETM | | one-/two-way |
|---|---|---|---|---|---|---|
| sea surface temperature | `t_seasfc` | [K] | $\Longleftarrow$ | `T(:,:,kmax)` | [°C] | (   2w) |
| mean sea level air pressure | `pres_msl` | [Pa] | $\Longrightarrow$ | `slp` | [Pa] | (1w, 2w) |
| gridscale rain rate | `rain_gsp_rate` | $\left[\mathrm{kg\,m^{-2}\,s^{-1}}\right]$ | | | | |
| gridscale snow rate | `snow_gsp_rate` | $\left[\mathrm{kg\,m^{-2}\,s^{-1}}\right]$ | | | | |
| gridscale graupel rate | `graupel_gsp_rate` | $\left[\mathrm{kg\,m^{-2}\,s^{-1}}\right]$ | | | | |
| gridscale hail rate | `hail_gsp_rate` | $\left[\mathrm{kg\,m^{-2}\,s^{-1}}\right]$ | $\Longrightarrow$ | `precip` | $\left[\mathrm{m\,s^{-1}}\right]$ | (1w, 2w) |
| gridscale ice rate | `ice_gsp_rate` | $\left[\mathrm{kg\,m^{-2}\,s^{-1}}\right]$ | | | | |
| convective rain rate | `rain_con_rate` | $\left[\mathrm{kg\,m^{-2}\,s^{-1}}\right]$ | | | | |
| convective snow rate | `snow_con_rate` | $\left[\mathrm{kg\,m^{-2}\,s^{-1}}\right]$ | | | | |
| surface moisture flux | `qhfl_s` | $\left[\mathrm{kg\,m^{-2}\,s^{-1}}\right]$ | $\Longrightarrow$ | `evap` | $\left[\mathrm{m\,s^{-1}}\right]$ | (1w, 2w) |
| u-momentum flux at surface | `umfl_s` | $\left[\mathrm{N\,m^{-2}}\right]$ | $\Longrightarrow$ (R) | `tausx` | $\left[\mathrm{N\,m^{-2}}\right]$ | (1w, 2w) |
| v-momentum flux at surface | `vmfl_s` | $\left[\mathrm{N\,m^{-2}}\right]$ | $\Longrightarrow$ (R) | `tausy` | $\left[\mathrm{N\,m^{-2}}\right]$ | (1w, 2w) |
| surface sensible heat flux | `shfl_s` | $\left[\mathrm{W\,m^{-2}}\right]$ | | | | |
| surface latent heat flux | `lhfl_s` | $\left[\mathrm{W\,m^{-2}}\right]$ | $\Longrightarrow$ | `shf` | $\left[\mathrm{W\,m^{-2}}\right]$ | (1w, 2w) |
| longwave net flux at surface | `thb_s` | $\left[\mathrm{W\,m^{-2}}\right]$ | | | | |
| shortwave net flux at surface | `sob_s` | $\left[\mathrm{W\,m^{-2}}\right]$ | $\Longrightarrow$ | `swr` | $\left[\mathrm{W\,m^{-2}}\right]$ | (1w, 2w) |
| zonal wind in $10\,\mathrm{m}$ | `u_10m` | $\left[\mathrm{m\,s^{-1}}\right]$ | $\Longrightarrow$ (R) | `u10` | $\left[\mathrm{m\,s^{-1}}\right]$ | |
| meridional wind in $10\,\mathrm{m}$ | `v_10m` | $\left[\mathrm{m\,s^{-1}}\right]$ | $\Longrightarrow$ (R) | `v10` | $\left[\mathrm{m\,s^{-1}}\right]$ | |
| temperature in $2\,\mathrm{m}$ | `t_2m` | [K] | $\Longrightarrow$ | `t2` | [K] | |
| dew point in $2\,\mathrm{m}$ | `td_2m` | [K] | $\Longrightarrow$ | `hum` | [K] | |
| relative humidity in $2\,\mathrm{m}$ | `rh_2m` | $\left[1 \times 10^{-2}\right]$ | $\Longrightarrow$ | `hum` | $\left[1 \times 10^{-2}\right]$ | |
| total cloud cover | `clct` | [1] | $\Longrightarrow$ | `tcc` | [1] | |

**Table 1.** List of quantities which can be exchanged in ICONGETM. The direction is indicated by the arrow. The units of the source and target variables are given in square brackets. Data conversion and aggregation is done automatically in the coupler. `precip` and `evap` are obtained by division with the reference density of fresh water. The corresponding contributions to precipitation from graupel, hail and ice are only considered for the coupling if they are activated in ICON. Wind data need to be rotated (R) to the local coordinate system in GETM. The exchanged humidity quantity (dew point or relative humidity) is correctly identified by the name attribute of the connected ESMF field. The possibility to exchange either flux data (3rd block) or state variables (last block) offers the comparison of different coupling strategies within the same model environment. The last column indicates which data are exchanged during the performed one- and two-way coupled simulations. The exchange of state variables is not applied in the simulations presented in Sec. 4.

`ESMF_Mesh` for structured and unstructured discretizations, respectively, see Sec. 3.1 and 3.2. Moreover, `ESMF_Fields` are created to advertise all data which are available for exchange. However, based on the user-specified lists of data that should be received by each model component, the model system automatically detects the required subset of fields which are finally connected and realized. The current implementation supports the exchange of flux and state data. The exchange of state variables is not applied in the simulations presented in Sec. 4 since only flux data are transferred. For a list of exchangeable quantities and their optional conversion by the mediator, see Tab. 1.

The data transfer between the NUOPC-Models via the NUOPC-Mediator is then prepared generically, i.e. by the NUOPC layer. NUOPC-Connectors are set up to redistribute the data between the different computing units used by the coupler and model components. For the actual regridding (interpolation) between the horizontal triangular grid from ICON and the horizontal latitude-longitude grid from GETM, one ESMF exchange grid (`ESMF_XGrid`) is created for each direction, for details see Sec. 3. The interpolation weights are calculated only once during initialization and are used in the Run phase. The generation of the `ESMF_XGrid` and the interpolation weights is the most expensive part of the overall overhead due to coupling. The later performed interpolation in the Run phase is relatively cheap.

In the present implementation, no model receives data during the Initialization phase. However, the first data exchange takes place at the beginning of the Run phase, as specified in the run sequence. All model components update their export fields at the end of the Initialization phase.

### 2.3.2 Run

During runtime the coupled model system is integrated in time by repeating the prescribed run sequence with the given coupling intervals until the simulation end time is reached. At the beginning of the run sequence, new input data are provided to each model component by data exchange and regridding via the mediator component. In ICON, the received data must be copied to model internal memory locations. For GETM, the `ESMF_Fields` already contain pointers to the internal memory. With the new data from the import fields, each model advances with its own time step until the next coupling time point is reached. At the end of the run sequence, all model components prepare the following data exchange by updating their export fields from the internal model memory.

### 2.3.3 Finalization

This phase finalizes all ESMF and NUOPC components. The finalization of the model compoents is included by calling the finalizing interface in ICON and GETM. The overall last step is the finalization of ESMF.

## 3 Data exchange between ICON and GETM

The data exchange between ICON and GETM is based on the regridding from the source model grid to an exchange grid and the regridding from the exchange grid to the target model grid. The ESMF exchange grid (`ESMF_XGrid`) infrastructure is used for the conservative interpolation at the air-sea interface, see the NUOPC-Mediator in Fig. 1. The aim is to apply

an interpolation approach which is independent of any horizontal resolution in ICON and GETM. Before the `ESMF_XGrid` and how it is utilized in ICONGETM is explained in detail, the horizontal discretization of ICON and GETM is presented. Furthermore, the interpolation is schematically described.

## 3.1 Triangular mesh in ICON

The horizontal grid structure of ICON is described in detail by Linardakis et al. (2011). The very first assumption for the horizontal grid is that the Earth is approximated as a sphere. It is based on the projection of an icosahedron onto the sphere. The edges of each triangle of the icosahedron can now be interpreted as an arc of great circles on the sphere. A refinement of the grid, i.e. to increase the resolution by using smaller triangles, is achieved by a combination of two steps. The first step is an initial division of the original icosahedron triangle edges by $n \in \mathbb{N}$. The second step are $k \in \mathbb{N}$ bisections of the remaining smaller triangles. The final grid is then described by R$n$B$k$. The number of triangles on the sphere for a grid R$n$B$k$ is given by $20n^2 4^k$, see Zängl et al. (2015). The effective grid resolution is given by

$$\sqrt{\frac{\pi}{5}} \frac{r_{\mathrm{E}}}{n2^k} \tag{1}$$

with Earth radius $r_{\mathrm{E}}$. Table 1 in Zängl et al. (2015) shows different R2B$k$ grids with effective grid resolutions. The DWD applies a global R3B07 grid, a R3B08 Europe-grid and a R3B09 Germany-grid for the weather forecast simulations, which have effective resolutions of $13.15\,\mathrm{km}$, $6.58\,\mathrm{km}$ and $3.29\,\mathrm{km}$, respectively.

The construction of refined grids supports a straight-forward nesting. An example for the Baltic Sea region based on R2B08, R2B09 and R2B10 grids with effective resolutions of $9.89\,\mathrm{km}$, $4.93\,\mathrm{km}$ and $2.47\,\mathrm{km}$ is shown in Fig. 2. Fig. 3 shows the R2B10 grid over the Island of Gotland in the Central Baltic Sea. Based on various external datasets (e.g. Reinert et al., 2020) every grid cell is associated with a set of fraction values for different land classifications, e.g. forest, urban areas and others. Cells with less than $50\,\%$ of land fraction are considered entirely as ocean cells, and vice versa. The triangular grid and the associated cell classification are stored in an `ESMF_Mesh` object, which also contains information about the domain decomposition onto computing units. The creation of the `ESMF_Mesh` is computing unit specific. Therefore, the domain distribution among the available computing units performed by ICON is kept in the `ESMF_Mesh` in ICONGETM.

## 3.2 Structured grid in GETM

The grid in GETM is structured and supports curvilinear horizontal coordinates in Cartesian and latitude-longitude space. For coupling with ICON, only grids in spherical coordinates can be used. A land mask defines land and water cells, see Fig. 3. Coordinate, area (defined through rhumb lines) and mask data as well as information about the domain distribution on computing units are stored in an `ESMF_Grid` object.

## 3.3 Exchange grid in the coupler

Based on the information provided by the mesh from ICON and the grid from GETM, an exchange grid is created in the coupler. The ESMF library constructs the exchange grid by overlaying both meshes, see Fig. 4, calculating the intersection

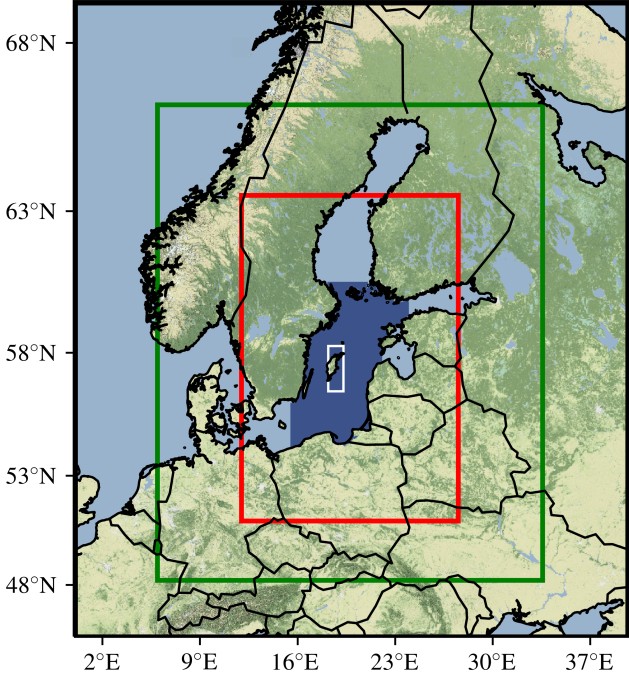

**Figure 2.** Nesting of different ICON domains with effective resolutions of $9.89\,\mathrm{km}$ (black frame), $4.93\,\mathrm{km}$ (green frame) and $2.47\,\mathrm{km}$ (red frame) over the Baltic Sea region. The darkblue area in the Central Baltic Sea represents the model domain of GETM. The white rectangle frames the area shown in Fig. 3.

points and conducting a final triangulation of all elements. For a schematic representation see Fig. 5. The `ESMF_XGrid` object only consists of elements that are required for the data exchange between the ocean cells in ICON and GETM. As indicated

in Figs. 4 and 5, the overlay of the different grids yields four possible combinations of land/ocean masks:

1. land cells in ICON and GETM,

2. ocean cell in ICON and land cell in GETM,

3. land cell in ICON and ocean cell in GETM,

4. ocean cells in ICON and GETM.

Elements of case 1 and 2 are excluded from the exchange grid, while elements of case 4 are included. Whether the elements of case 3 belong to the exchange grid depends on the direction of interpolation. Therefore, two different exchange grids are created and used: one for the interpolation from ICON to GETM, which includes the elements of case 3, and one vice versa, excluding elements of case 3, see Fig. 6.

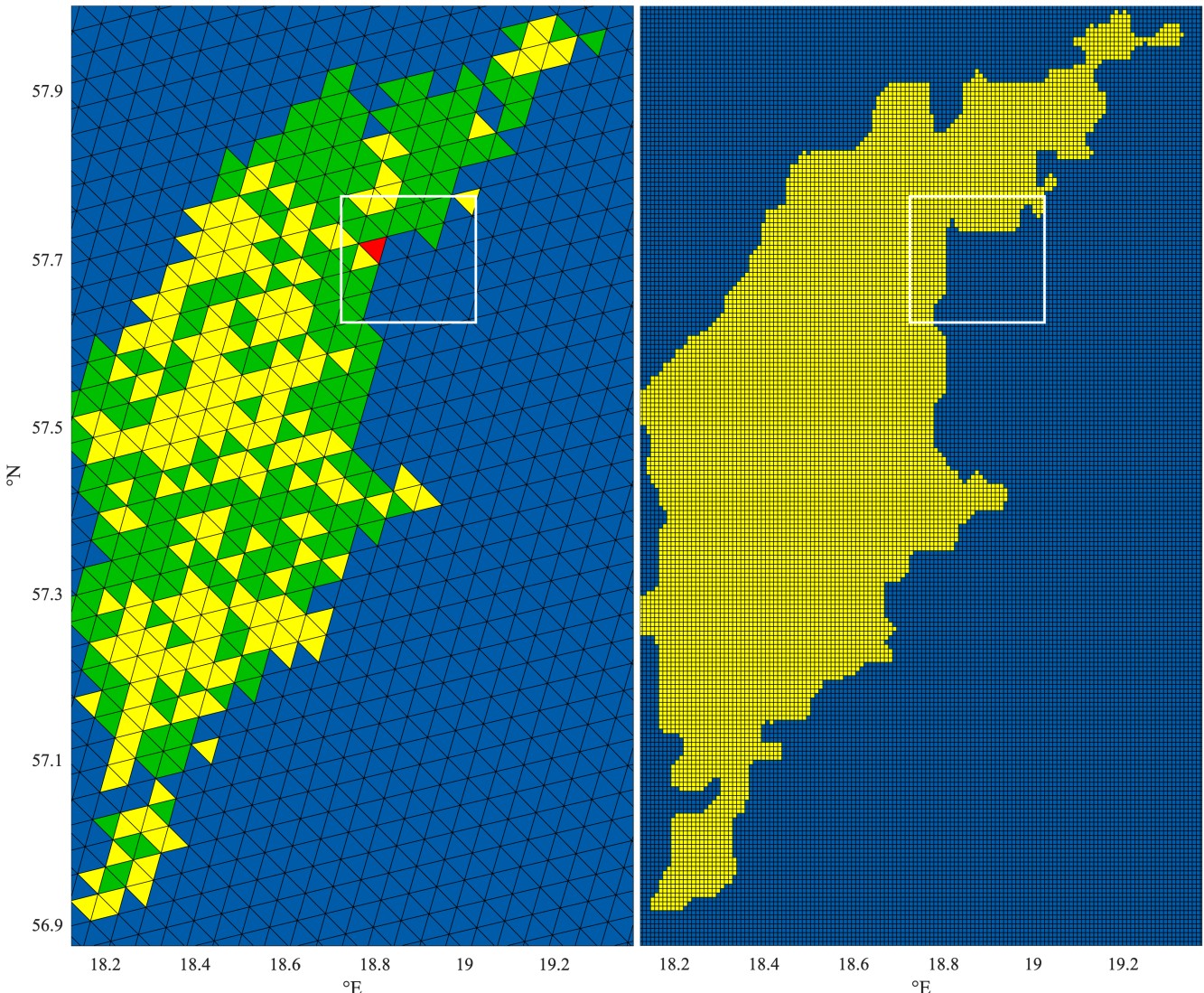

**Figure 3.** Triangular grid with an effective resolution of $2.47\,\mathrm{km}$ used in ICON (left) and rectangular grid with a resolution of approximately $600\,\mathrm{m}$ used in GETM (right) over the Island of Gotland in the Central Baltic Sea, see Fig. 2. In the ICON grid, the different colouring represents cells that consist of more than 50% of ocean (blue), forest (green), urban areas (red) or non-specific land classifications (yellow). GETM only distinguishes between ocean (blue) and land (yellow). The white rectangles frame the area shown in Fig. 4.

## 3.4 Regridding

One major challenge for the coupling between the unstructured grid of ICON and the structured grid of GETM is the interpolation of data on scattered nodes. The irregularity of the unstructured grid complicates the selection of the stencil. The correct interpolation weights for a conservative interpolation require the determination of the intersections of the source and target

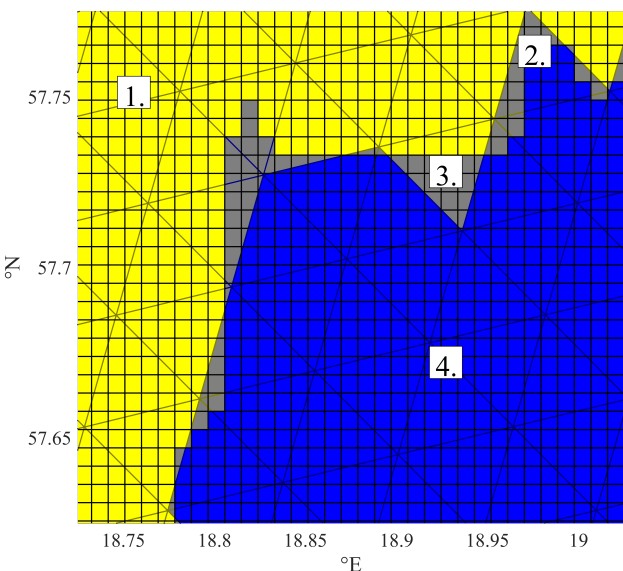

**Figure 4.** Overlay of the triangular ICON grid and the rectangular GETM grid at the eastern coast of the Island of Gotland in the Central Baltic Sea, see Fig. 3. The four possible combinations of land/ocean masks are labeled. Gray areas mark different land/ocean masks: ICON ocean and GETM land (case 2), ICON land and GETM ocean (case 3).

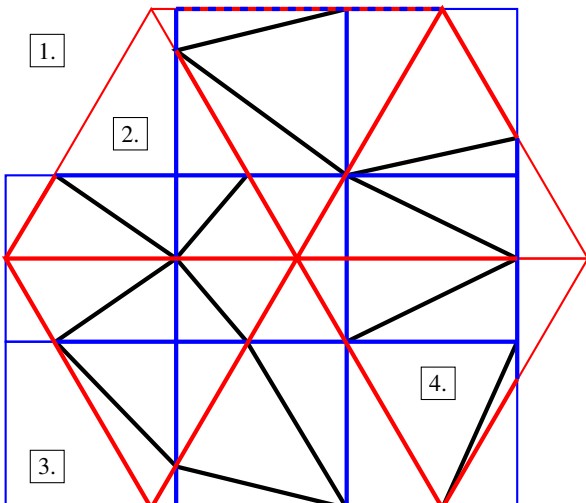

**Figure 5.** Exemplary 2D exchange grid formed by a triangular atmosphere (red) and a rectangular ocean (blue) grid. The exchange grid consists of edges from the original triangular and rectangular grids (thick red and blue) and additional edges from the triangulation (black). Assuming that only water cells are shown, the four possible combinations of land/ocean masks are labeled. Here the exchange grid is shown for the interpolation from the ocean to the atmosphere grid, therefore, excluding the elements of case 3.

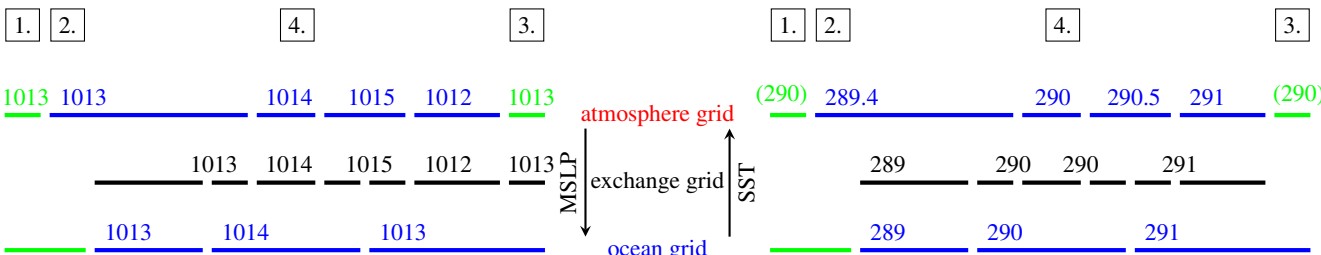

**Figure 6.** Schematic representation of the regridding between ICON and GETM. In the atmosphere and ocean grids active ocean cells are coloured in blue and land cells in green. As shown for the transfer of mean sea level pressure (MSLP in hPa) and sea surface temperature (SST in K), the exchange grid can consist of different cells for each direction. The four possible combinations of land/ocean masks are indicated. On land (cases 1 and 3) an ICON-internal SST (here 290 K) is used. This ICON-internal SST is also considered for fractions of ocean cells not covered by GETM ocean cells (case 2).

grids, and the calculation of the resulting areas. The processing of distributed neighbor information in unstructured grids also requires performant data structures and algorithms. The ESMF exchange grid (`ESMF_XGrid`) and the associated interpola-
tion weights stored in the `ESMF_RouteHandle` hide all these aspects from the user and provide an efficient and automatic conservative interpolation infrastructure.

The `ESMF_XGrid` class supports first and second order conservative interpolation. Currently, only the first order method has been applied in ICONGETM. The interpolation weights are calculated during the initialization, based on the areas of the grid cells. The connecting edges between the vertices in the exchange grid are defined on arcs of great circles, which differ from
the rhumb lines used in GETM. However, the interpolation between GETM and the exchange grid is still conservative, because the weights are scaled in terms of the area provided by GETM.

### 3.4.1 Regridding from ICON to GETM

As sketched in Fig. 6, the interpolation of the mean sea level pressure (MSLP) from ICON to GETM is straight-forward in principle, because ICON provides all quantities over the whole domain. However, in case sea surface fluxes are exchanged,
there are two issues if the land/sea masks do not match between ICON and GETM. First, there is a physical inconsistency, when surface fluxes parametrized over land cells in ICON are transferred to ocean cells in GETM (case 3). Second, when ICON applies sea surface fluxes in ocean areas that are represented by land in GETM (case 2), the fluxes are not conserved in the global atmosphere-ocean-system. This latter case demonstrates that the conservative interpolation via the exchange grid is not sufficient to guarantee a conservative flux exchange.

 ### 3.4.2 Regridding from GETM to ICON

Fig. 6 sketches the regridding of the sea surface temperature (SST). The update of an ICON ocean cell that is partly covered by a GETM land cell (case 2) needs some remarks. For the contribution from a GETM land cell to an ICON ocean cell, the SST value of the ICON cell is applied. This value can be either a user-provided ICON-internal SST, if the climatological update is activated, or simply the SST from the last time step. For the first time step this is the initial ICON-internal SST.

## 4 Demonstration

For demonstration purposes, the newly developed model system ICONGETM is applied to the Central Baltic Sea. High-resolution uncoupled, one-way and two-way coupled simulations are carried out and compared. The modelling period July 1 – 21, 2012 is chosen to evaluate the model results with measurement data from a field campaign with research vessel (RV) Meteor (cruise M87).

### 4.1 Coupled Central Baltic Sea setup

#### 4.1.1 ICON configuration

The ICON setup is based on the operational non-hydrostatic numerical weather prediction configuration from the German Weather Service (DWD), but covers a different model domain. For the coupled Baltic Sea setup, ICON is run in limited area mode with three nested domains with effective resolutions of $9.89\,\mathrm{km}$, $4.93\,\mathrm{km}$ and $2.47\,\mathrm{km}$, respectively, see Fig. 2. The vertical terrain-following hybrid grids consist of 90, 65 and 54 height-based vertical levels. The heights are pre-defined depending on the associated pressure in the US 1976 standard atmosphere, with the top boundary of the model domain depending on the numbers of levels (Reinert et al., 2020, Fig. 3.5). At the open boundaries, the outermost domain is driven by 6-hourly IFS data from ECMWF. The designed nesting guarantees a smooth transition from this coarse boundary forcing, provided with $16\,\mathrm{km}$ resolution, to the innermost domain over the Central Baltic Sea. The feedback from refined nesting levels is relaxation-based. The model time steps are $60\,\mathrm{s}$, $30\,\mathrm{s}$ and $15\,\mathrm{s}$, respectively. For all domains, initial conditions are obtained by interpolation from IFS data. In contrast to long term hindcast applications, ICON is not re-initialized during the model run. Within this "free run" the ICON-internal sea surface temperature, prescribed by the OSTIA data from the DWD (Donlon et al., 2012) with a resolution of $\frac{1}{20}^{\circ}$ (approx. $5\,\mathrm{km}$), is not updated by daily or monthly climatological increments.

The settings include also the sub-grid scale cloud scheme as well as the vertical diffusion and transfer turbulent coefficients from COSMO. For the performed summer simulations, COSMO microphysics (Bechtold et al., 2008; Doms et al., 2011; Zängl et al., 2015) with only two frozen water substances (cloud ice and snow) are applied. The Rapid Radiation Transfer Model (RRTM) of Mlawer et al. (1997) is used. The convection parameterization is switched off for the finest resolved domain.

The complete configuration can be found in the code. The run scripts include the namelist settings. A detailed description of the namelist options are provided through the ICON documentation which is part of the ICON model code.

ICON does not need any specific settings when run two-way coupled in ICONGETM, because the coupler will simply over-
write the ICON-internal sea surface temperature with the data provided from GETM.

### 4.1.2 GETM configuration

The GETM setup for the Central Baltic Sea is taken from Holtermann et al. (2014). The model domain is shown in Fig. 2. Based
on an equidistant spherical grid, the horizontal resolution varies between $500\,\text{m}$ and $600\,\text{m}$. In the vertical, 100 terrain-following
layers with adaptive zooming towards stratification are applied. At the open boundaries, hourly data for temperature, salinity,
sea surface elevation and normal depth-averaged velocity from the Baltic Sea setup of Gräwe et al. (2019) are prescribed.
Furthermore, the freshwater discharge of the five major rivers entering the model domain is prescribed, see Chrysagi et al.
(2021) for details. The initial temperature and salinity distribution for the present study was obtained by continuing the original
simulations of Holtermann et al. (2014) and subsequent distance-weighted nudging with available measurements from the
HELCOM database (www.helcom.fi) below $50\,\text{m}$ depths. The 3D model time step is $45\,\text{s}$.

During a spin-up period from 20 May – 30 June 2012, GETM is run uncoupled. In the GETM configuration file, two namelist
parameters have to be changed for the uncoupled and coupled simulations. The first one specifies whether atmospheric data
should be read from file or whether an external coupler will take care of the data provision. A second one specifies whether
GETM needs to compute the air-sea fluxes during runtime or whether air-sea fluxes are already provided. In the uncoupled
simulation, GETM calculates the air-sea fluxes according to the bulk parameterization of Kondo (1975) in terms of hourly
meteorological CFSv2 data (Saha et al., 2014) read from file. During the one- and two-way coupled simulations the coupler
will process the air-sea fluxes from ICON.

### 4.1.3 ICONGETM configuration

The exchanged data for the one- and two-way coupled simulations are listed in Tab. 1. In order to temporally resolve the fast
feedbacks between atmosphere and ocean dynamics, the coupling time step is set to three minutes, the least common multiplier
of the time steps from ICON and GETM. For the present setup, a good concurrent load-balancing with minimum idle/waiting
times for ICON and GETM was empirically obtained through the log-file time information resulting in 864 and 384 processes,
respectively.

## 4.2 Results

### 4.2.1 Effects of interactive coupling on meteorology

In the uncoupled and one-way coupled simulations, ICON uses its prescribed internal sea surface temperature (SST), which
does not show any pronounced temperature gradients due to oceanic eddies or coastal upwelling. Short-term and small-scale
variations are only considered in the two-way coupled ICONGETM run, see Fig. 7, with the SST simulated and provided in
high-resolution by GETM.

In July 2012, the simulated SST ranged around $289\,\text{K}$, with values below $282\,\text{K}$ in the upwelling areas south of the coast of

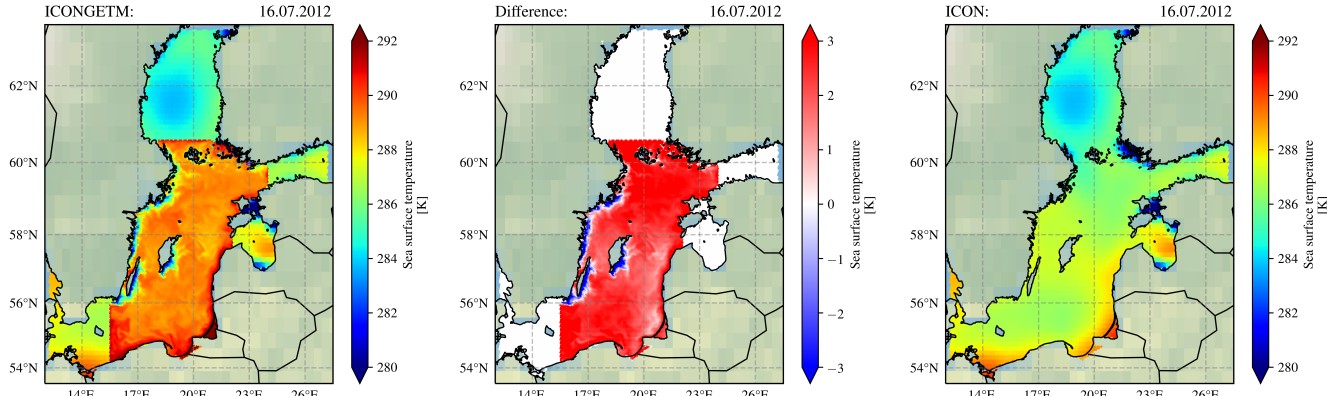

**Figure 7.** Daily mean sea surface temperature (SST) from the two-way coupled ICONGETM run (left panel), and the uncoupled/one-way coupled ICON run (right panel), as well as the difference (central panel; ICONGETM minus ICON) for 16 July 2012. Outside the domain of simulated SST in the Central Baltic Sea, the two-way coupled ICONGETM run also uses the prescribed ICON-internal SST.

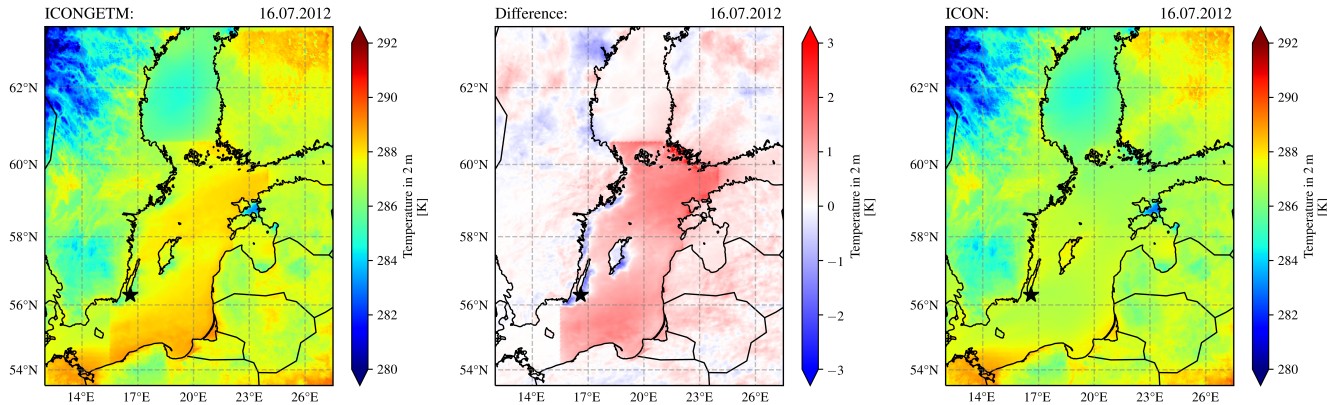

**Figure 8.** Daily mean 2 m air temperature from the two-way coupled ICONGETM simulation (left panel) and the uncoupled/one-way coupled ICON simulation (right panel), as well as the difference (central panel; ICONGETM minus ICON) for 16 July 2012. The black star south-east of the island of Öland marks the position of the vertical profiles shown in Fig. 12.

mainland Sweden and the islands of Öland and Gotland. The ICON-internal SST is between 0.5 K and 2 K colder. The overall warmer surface of the Baltic Sea in the two-way coupled ICONGETM run causes a predominantly warmer lower troposphere. As a result, the daily mean 2 m temperature is about 0.5 K to 2 K higher, cf. Fig. 8.

Over the upwelling regions, however, where cold deep water has risen to the surface, only the two-way coupled ICONGETM run is able to reproduce the cooling in the 2 m temperatures of between minus 1 K to 2 K against the surroundings. Thus, the two-way coupled atmosphere-ocean simulation provides a more realistic representation of actual weather conditions. This is also reflected when comparing the model results with air temperature measured onboard the RV Meteor off the island of

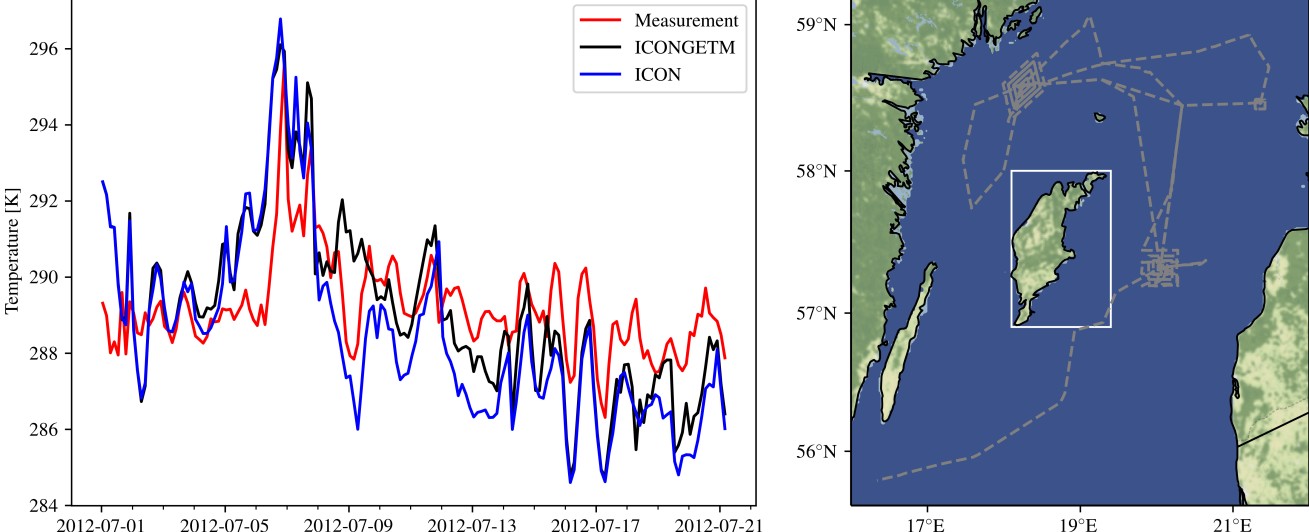

**Figure 9.** Air temperature in the Central Baltic Sea over the period July 1 – 21, 2012 (left panel). Compared are 3-hourly measurements in 29.1 m onboard the RV Meteor, ship track shown on the right panel, with model results from the two-way coupled ICONGETM and uncoupled/one-way coupled ICON simulations, respectively. The white frame shows the island of Gotland, similar to Fig. 2.

Gotland during the above-mentioned field campaign, see Fig. 9. In the uncoupled ICON simulation the temperature is signifi-
cantly underestimated by up to 2.5 K. In contrast, the values from the two-way coupled ICONGETM run are in a much better
agreement with the measurements. The temporal development agrees also more with the observations, see Fig. 9, especially
after 10 days of simulations. The average deviation between the modelled and measured temperature in the period from 01 July
till 21 July 2012 is decreased from 1.9 K for the uncoupled to 1.6 K for the two-way coupled simulation. This represents an
improvement of about 15 %. On the other hand, the Pearson correlation coefficient is only slightly improved from 0.7 for the
uncoupled to 0.72 for the two-way coupled simulation. Fig. 9 indicates that the coupled ICONGETM system needs some spin-
up time to adapt to the coupling, before the improvement with respect to the uncoupled simulation becomes visible. Within the
period from 10 July till 21 July 2012, the average deviation between the modelled and measured temperature decreases from
2.0 K for the uncoupled to 1.5 K for the two-way coupled simulation. Thus, after the spin-up, the model results are significantly
improved due to the coupling by 25 %. The removal of the spin-up period also increases the correlation coefficients to 0.73 for
the uncoupled and to 0.75 for the two-way coupled simulation.
The interactive coupling between ICON and GETM also affects the synoptic-scale dynamic meteorology and leads to local
effects in the atmospheric boundary layer. The warmer Baltic Sea and higher lower-troposphere temperatures in the two-way
coupled ICONGETM simulation result in a mean sea-level pressure that is up to 1 hPa lower over sea and adjacent land than
in the uncoupled/one-way coupled ICON run, cf. Fig. 10.
  Therefore, the low-pressure area over the northern Baltic Sea, which causes the observed upwelling event, is even stronger

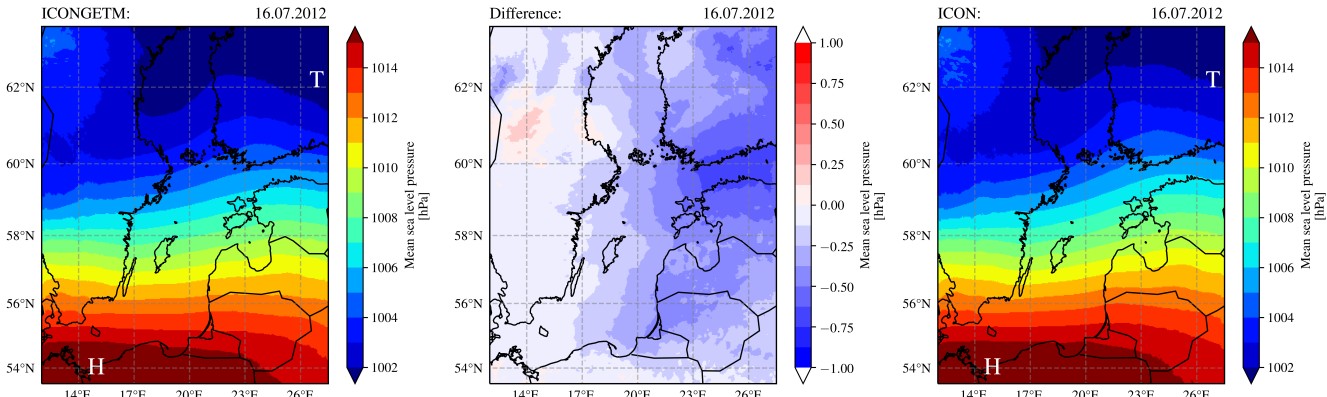

**Figure 10.** Daily mean sea-level pressure from the two-way coupled ICONGETM simulation (left panel) and the uncoupled/one-way coupled ICON simulation (right panel), as well as the difference (central panel; ICONGETM minus ICON) for 16 July 2012. 'T' and 'H' mark surface lows and highs, respectively.

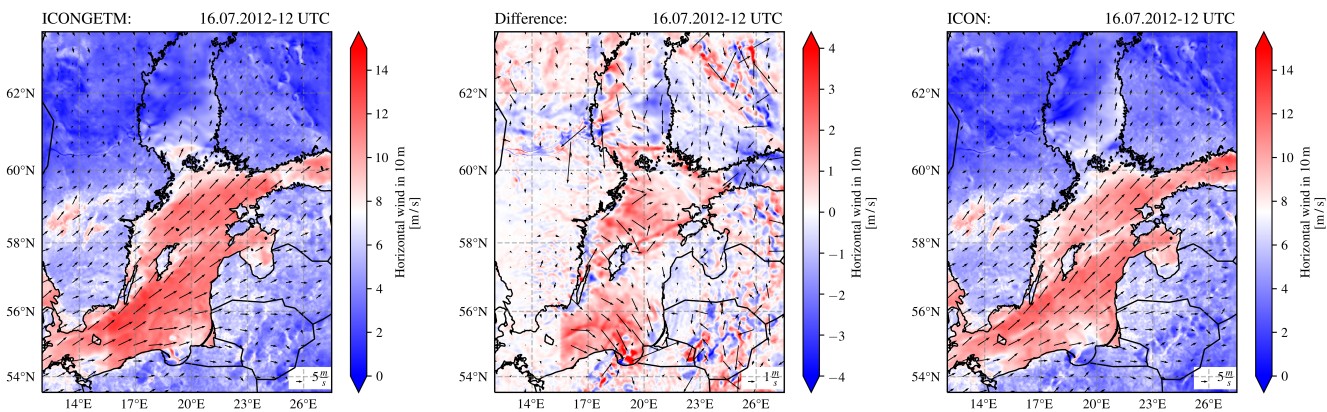

**Figure 11.** 10 m horizontal wind field from the two-way coupled ICONGETM simulation (left panel) and the uncoupled/one-way coupled ICON simulation (right panel), as well as the difference (central panel; ICONGETM minus ICON) for 16 July 2012 12 UTC. Displayed are the wind vectors (reference vector at the bottom of the figure, units of $\mathrm{m\,s^{-1}}$) and the wind speed (coloured).

in the two-way coupled simulation. The resulting higher pressure gradient between the Baltic low and the high over Western Europe, cf. Fig. 10, leads to an increase of the near-surface wind field over a large part of the water surface, while locally wind velocity is reduced in the upwelling regions, see Fig. 11.

The weather conditions leading to the upwelling event are therefore more pronounced in the two-way coupled model run. The effects of the interactive atmosphere-ocean coupling on the boundary layer dynamics is most evident for the upwelling regions. Fig. 12 shows vertical profiles of potential temperature and specific humidity over the upwelling area east of Öland, see star marker in Fig. 8.

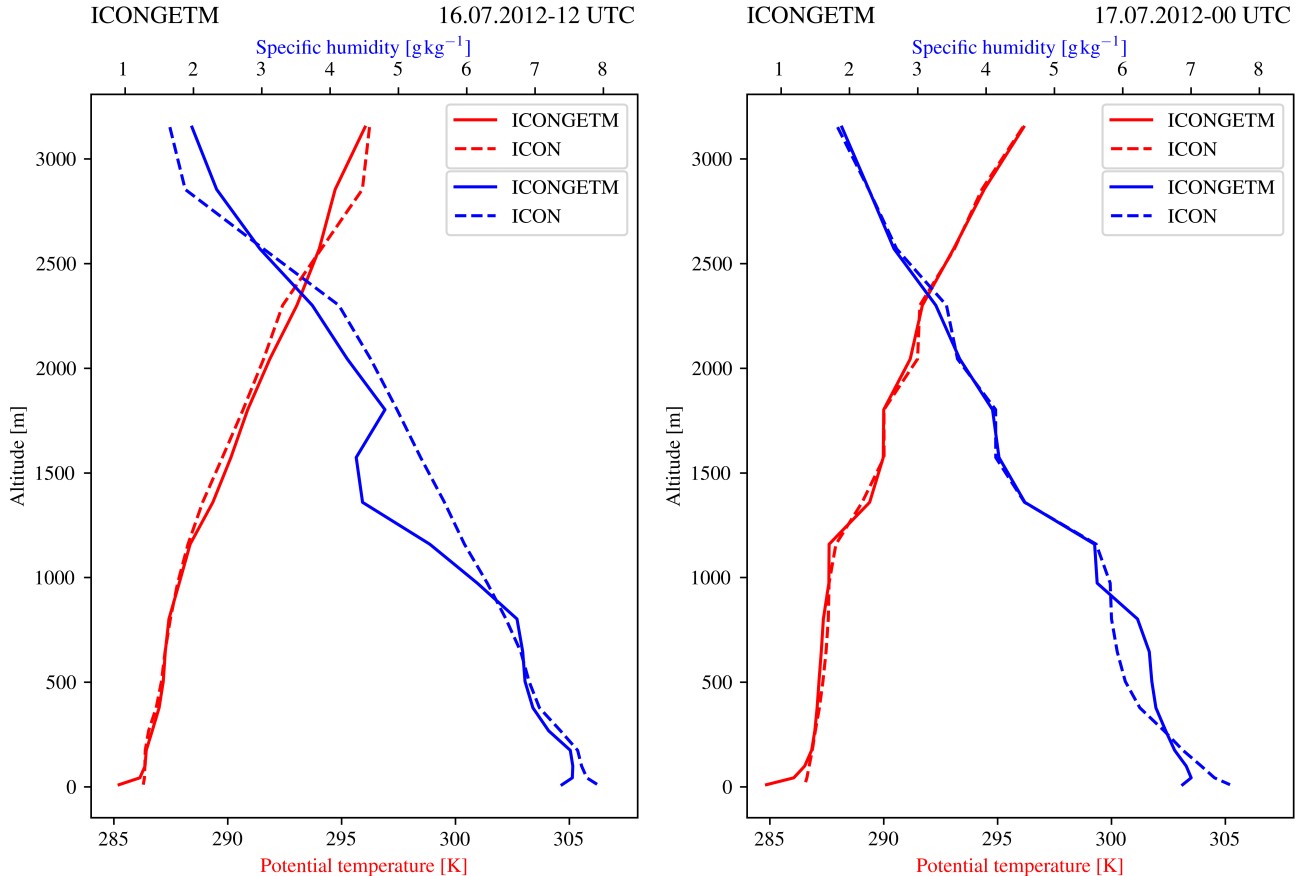

**Figure 12.** Atmospheric vertical profiles of potential temperature and specific humidity from the two-way coupled ICONGETM run and the uncoupled/one-way coupled ICON run, for 16 (left) and 17 (right) July 2012 at 12 UTC and 00 UTC, respectively. The profiles are obtained south-east of the island of Öland, see black star in Fig. 8.

Compared are the profiles for 16 July 2012 at noon and midnight, when the upwelling event was most pronounced in this area. As a result of the upwelling of cold deep water, the potential temperature is reduced by up to $1.5\,\mathrm{K}$ to $2\,\mathrm{K}$ and atmospheric stratification is increased in the lowermost $50\,\mathrm{m}$ to $150\,\mathrm{m}$ at noon and mid-night, respectively. The two-way coupled ICONGETM run also shows slightly enhanced gradients in the potential temperature profile at the upper boundary layer. The more stable stratification has an effect on the boundary-layer mixing, whereby humid air is more concentrated in the central to upper part of the boundary layer, between $900\,\mathrm{m}$ and $2400\,\mathrm{m}$ in left panel of Fig. 12. Due to reduced evaporation, it is less in the lowermost part, below $500\,\mathrm{m}$ in left panel of Fig. 12. In addition, there is less momentum mixed downwards (not shown), which is a likely explanation for the locally reduced wind velocity in the upwelling regions at Sweden's mainland coast and the Öland and Gotland islands, shown by negative differences in the central panel of Fig. 11. In the coupled case, the temperature

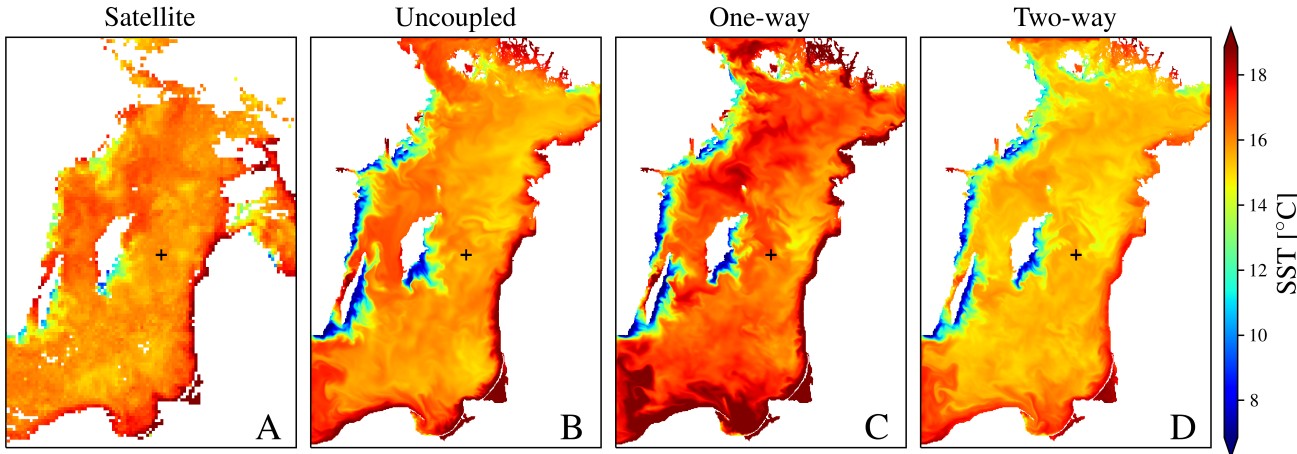

16.07.2012

Satellite    Uncoupled    One-way    Two-way

**Figure 13.** Daily mean sea surface temperature from satellites (A) and simulated by GETM in the uncoupled (B), one-way (C) and two-way (D) coupled simulation for 16 July 2012. The colorbar is identical to Fig. 7. The SST derived from satellite data was provided by the Federal Maritime and Hydrographic Agency of Germany (BSH). The black cross marks the position of station TF271 in the Eastern Gotland Basin.

gradient between land and sea is increased in the area of the upwelling, cf. Fig. 8, with almost the same land temperatures but significantly lower SSTs, which locally increases the onshore wind component and thus weakens the overall more easterly wind in Fig. 11.

Hence, the evolution/stratification of the marine boundary layer are reproduced more realistically. Similarly, also the boundary
layer wind conditions, in particular over upwelling regions, are better represented using two-way atmosphere-ocean coupling. The wind stress being coupled with the SST is largely related to atmospheric stability effects rather than to the change of the surface wind speed. This has also been shown recently in Fallmann et al. (2019).

### 4.2.2   Coupling effects in the ocean

In Fig. 13, the sea surface temperature (SST) from all model simulations are compared to satellite data. Due to the forcing
with meteorological reanalysis data, the SST from the uncoupled simulation shows best agreement with the satellite data and most pronounced upwelling activity. The SST from the two-way coupled simulation is only slightly colder, but is clearly overestimated in the one-way coupled simulation. This overestimation results from a continuous increase of near surface temperature, see Fig. 14 for the evolution in the Eastern Gotland Basin.

The evolution indicates that the surface heat flux (not shown) used in the one-way coupled GETM simulation is overestimated
after 12 July 2012. For the one-way coupled simulation, the heat flux provided by ICON is calculated in terms of the too cold ICON-internal SST, see Fig. 7. In the uncoupled and two-way coupled simulations, the surface heat flux is calculated in terms of the SST from GETM, either within GETM or ICON, respectively. Henceforth, the fluxes are adapting more conveniently

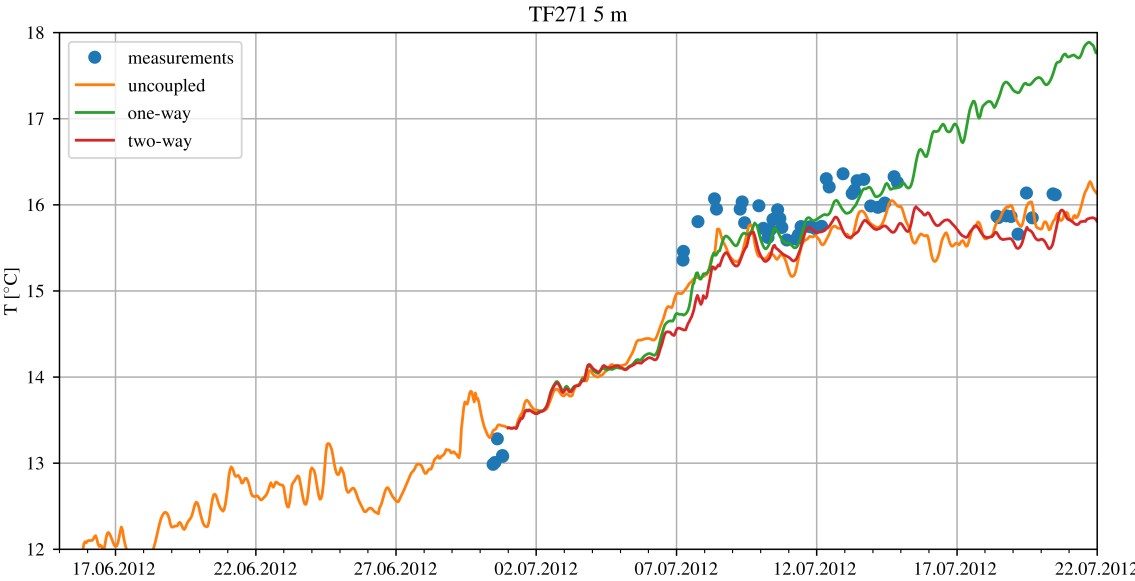

**Figure 14.** Temperature in 5 m depth at station TF271 from CTD measurements and the three model simulations. The one- and two-way coupled simulations are started at 1 July 2012, after the uncoupled spin-up period.

to the warming ocean. The temperature differences are not only confined to the sea surface, see Fig. 15 for vertical profiles of temperature and salinity in the Eastern Gotland Basin.

In the upper $20\,\mathrm{m}$, the temperatures from the uncoupled and two-way coupled simulations are very similar and do excellently agree with the measurements, cf. Fig. 15 B. The temperature from the one-way coupled simulation is approximately $1.5\,\mathrm{K}$ too warm. Within the thermocline, $20-40\,\mathrm{m}$ depth, the temperature profiles do show a stronger difference. When these deviations are compared against the temporal variability of the temperature in an 8 days interval, it becomes clear that the differences can be attributed to the natural variability of the thermocline in the Central Baltic Sea, see Fig. 15 B. For a better visibility, only
the variability of the uncoupled simulation is shown. A slightly different excitation timing of wind driven processes, i.e. near inertial internal waves, are subsequently causing the differences between the analysed profiles.

The salinity differences between the simulations show, in analogy to the temperature, deviations in the thermocline, but are also within the variability observed over an 8 day time period, see Fig. 15 D. In contrast to the surface, the deep water below the thermocline is virtually not affected by the different atmospheric forcing, see Fig. 15 A and C, which is due the strong density
gradients in the thermo- and halocline, inhibiting a significant turbulent transport of heat and salt on the timescales analysed here (Reissmann et al., 2009; Holtermann et al., 2020).

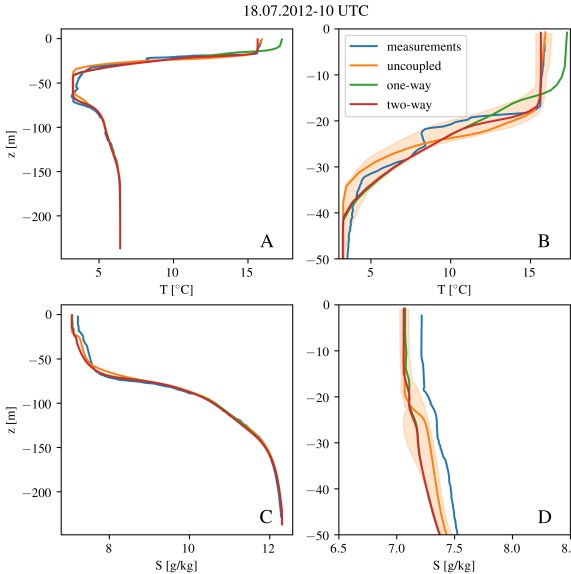

**Figure 15.** Temperature and salinity profiles at station TF271 from CTD measurements and the three model simulations. Panels A and C depict the whole water column, B and D a zoom towards the sea surface. The light orange shaded area depicts the variability of the uncoupled simulation within an 8 days time interval (14 – 21 July 2012).

## 5 Discussion

ICONGETM supports the exchange of fluxes and state variables across the air-sea interface. The applied ESMF exchange grids guarantee a conservative flux exchange, except in the area where the land/sea masks of ICON and GETM do not match. The

NUOPC-Mediator performs additional unit conversion and merging of precipitation fluxes, see Tab. 1. In ICONGETM v1.0, the air-sea fluxes are taken from the atmosphere model ICON. Their calculation in ICON is very complex and deeply nested in the model core.

Ideally all fluxes, air-sea and land fluxes, should be calculated directly on one unique ESMF exchange grid in the mediator and applied as boundary conditions to the corresponding individual models (Best et al., 2004). On the exchange grid, a unique

land/sea mask of the coupled system can be defined. If the land/sea mask of the exchange grid is adjusted to the mask of the ocean model (Balaji et al., 2006), the associated sea surface areas will be identical. In this case, the conservative interpolation between the exchange and model grids finally offers a fully conservative flux exchange between the atmosphere and ocean, despite originally non-matching land/sea masks in the individual models. Moreover, physical consistency will be ensured in the sense that only air-sea fluxes, i.e. fluxes influenced by the sea surface temperature, are provided to the ocean. If the sea surface

area on the exchange grid does not cover the one of the ocean model, creep, nearest neighbour or other extrapolation methods are required to avoid the application of land fluxes to the ocean (see e.g. Kara et al., 2007; Chen et al., 2010; Turuncoglu,

2019). In any case, fluxes provided by the mediator can be applied in the atmosphere and ocean over the same period until new fluxes are calculated in the next coupling time step. The flux calculation on the ESMF exchange grid in a central mediator component also offers the most straight-forward extension of the coupled system by models for e.g. waves and sea ice. One

drawback of the flux calculation outside the individual models can be stability issues for explicit time stepping schemes or complex coupling implementations for implicit time stepping schemes.

Alternatively, a conservative exchange of air-sea fluxes calculated in the atmosphere model is possible, if the mask of the exchange grid can be emulated in the atmosphere model due to mixed land/ocean cells. For this, the water fraction in each cell must be obtained by conservative interpolation of the sea surface area from the ocean model via the ESMF exchange grid.

In its present state, ICON does neither support the described ideal modular coupling nor the alternative. Both approaches require non-trivial modifications to the ICON code. It is expected that they will become available in future releases of ICON, such that the full potential of ICONGETM for a flexible conservative flux exchange via the ESMF exchange grid can be exploited.

The two-way coupled simulation presented in the previous section was conducted with a coupling time step of $3\,\mathrm{min}$ and

showed an overhead of approximately $15\,\%$ compared to the uncoupled simulation. The majority is spent for the initialization. This demonstrates the excellent performance of the developed model system based on ESMF/NUOPC and its potential for future high-resolution coupled atmosphere-ocean simulations with fast feedback integration.

## 6    Conclusions

With ICONGETM, consisting of the state-of-the-art operational atmosphere model ICON and the established coastal ocean

model GETM, a new model system especially suited for high-resolution studies has been developed. The two-way coupled model system is driven by latest NUOPC coupling technology. The data exchange between the unstructured grid of ICON and the structured grid of GETM is carried out via a central mediator component. In contrast to other model systems with interpolation directly between model grids, the new implementation and usage of ESMF exchange grids in the mediator component has been described in detail. The added value and the potential of ESMF exchange grids for conservative interpolation,

flux calculations and coherent land/sea masks has been discussed. The functioning and performance of ICONGETM has been demonstrated. Thanks to NUOPC, future extensions of the model system by wave or sea ice models require only minimal implementational effort.

*Code availability.* The source code of ICONGETM is available from https://gitlab.com/modellers-tropos/icongetm.git under GPL license. A frozen version of the code and run scripts as used in this paper is archived on Zenodo (https://doi.org/10.5281/zenodo.4516568). The

modified source code of ICON is also archived on Zenodo (https://doi.org/10.5281/zenodo.4432739), and available if a valid Software License Agreement, obtained via https://code.mpimet.mpg.de/projects/icon-license, is presented to the first author. The source code of GETM was not modified and is available from https://sourceforge.net/p/getm/code/ci/iow/tree/ under GPL license.

## Appendix A:  List of model acronyms

| | |
|---|---|
| **ADCIRC** | ADvanced CIRCulation |
| **COAMPS** | Coupled Ocean/Atmosphere Mesoscale Prediction System |
| **COAWST** | Coupled-Ocean-Atmosphere-Wave-Sediment Transport modeling system |
| **COSMO** | Consortium for Small-scale Modelling |
| **ESMF** | Earth System Modeling Framework |
| **ESPS** | Earth System Prediction Suite |
| **FABM** | Framework for Aquatic Biogeochemical Models |
| **GETM** | General Estuarine Transport Model |
| **GOTM** | General Ocean Turbulence Model |
| **ICON** | ICOsahedral Non-hydrostatic modelling framework |
| **ICON-ESM** | ICON Earth system model |
| **ICONGETM** | ICON coupled with GETM |
| **IFS** | Integrated Forecast System |
| **MCT** | Model Coupling Toolkit |
| **MetUM** | Met Office Unified Model |
| **MITgcm** | MIT General Circulation Model |
| **NCOM** | Navy Coastal Ocean Model |
| **NEMO** | Nucleus for European Modelling of the Ocean |
| **NEMS** | NOAA Environmental Modeling System |
| **NUOPC** | National Unified Operational Prediction Capability |
| **OASIS** | Ocean Atmosphere Sea Ice Soil |
| **RegESM** | Regional Earth System Model |
| **ROMS** | Regional Ocean Modeling System |
| **RRTM** | Rapid Radiation Transfer Model |
| **SKRIPS** | Scripps–KAUST Regional Integrated Prediction System |
| **SWAN** | Simulating WAves Nearshore |
| **UKC2** | UK environmental prediction system |
| **WAVEWATCH III** | wave modeling framework |
| **WRF** | Weather Research and Forecasting model |
| **YAC** | Yet Another Coupler |

*Author contributions.* The code has been designed, developed and implemented by TPB in cooperation with KK. The demonstration setup was provided by TPB and BH for ICON and PH and KK for GETM. The coupling configuration has been prepared by TPB and KK and discussed with all authors. BH and TPB evaluated the meteorological results of the simulation, i.e. ICON. PH and KK evaluated the simulation results on the ocean side, i.e. GETM. HR advised and discussed the flux exchange as well as the coupling strategy. OK advised the code development and supported the implementation of mathematical utility routines. All authors contributed to this paper in the sections corresponding to their part during the work flow.

*Competing interests.* The authors declare having no conflict of interest.

*Acknowledgements.* This work is the outcome of the model coupling initiative LOCUS (Land-ocean interaction mediated by coastal upwelling and sea breeze) funded by the Leibniz institutes TROPOS and IOW.

The authors gratefully acknowledge additional financial support by the German Research Foundation for the Collaborative Research Center TRR181 on Energy Transfers in Atmosphere and Ocean (Project 274762653) and the funding of Peter Holtermann by grant HO 5891/1-1, as well as the project MOSSCO (Modular System for Shelves and Coasts; FKZ 03F0740B, funded by the German Federal Ministry of Research and Education). Furthermore, the authors would like to acknowledge Fei Lui, Robert Oehmke, Rocky Dunlap and Gerhard Theurich from the ESMF support, who replied to every support request and continuously improved the ESMF library. KK is thankful for the inspiring collaboration and discussions with Carsten Lemmen (Helmholtz Zentrum Geesthacht).

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
