# Peer review of "ICONGETM v1.0 – Flexible NUOPC-driven two-way coupling via ESMF exchange grids between the unstructured-grid atmosphere model ICON and the structured-grid coastal ocean model GETM"

_Geoscientific Model Development, 2020_

## Referee Comment (RC1) · Anonymous Referee #1 · 7 Nov 2020

This manuscript documents the coupling of an unstructured-grid atmospheric model (ICON, configured as a limited-area model) with a structured-grid coastal ocean model (GETM). It clearly describes the technical route and the model simulations. The utilization of a community-based coupler (NUOPC/ESMF) is a good example for other people who have similar interests. I believe that this work fits within the scope of GMD and deserves publication. My major concern is about its scientific quality. While I think GMD appreciates technical work and interdisciplinarity, the current manuscript (as a model description paper) does not offer enough information that could be useful to

the general readers. The major conclusion merely summarized what the authors have done: "The demonstration example shows that there is now a coupled model available which allows the investigation of processes at the air-sea interface with high-resolved model simulations." I do believe that the manuscript offers more than that, and it can be further improved. I have some questions and comments which might be helpful to the authors.

1. I find one useful aspect of this manuscript is to offer an example of coupling an unstructured-grid atmospheric model with a structured-grid ocean model, based on a community coupler (NUOPC/ESMF). It would be valuable to put the current work into a broader background. Is there any earlier study that has already explored along this line (including global and regional configuration)? If so, the authors should give a general overview; if not, the present work would be more unique and the authors should explicitly speak out.

2. What is the major challenge of coupling an unstructured-grid atmospheric model with a structured-grid coastal ocean model? Or more general, any unstructured-grid model (atmosphere/ocean) with a structured-grid model.

3. What is the unique aspect of using NUOPC for this particular work? In comparison with other community-based couplers such as OASIS. It's also useful to briefly review the existing coupled models based on NUOPC.

4. It would be useful to give more details on the construction of a coupled model within NUOPC, for instance, showing some prototype codes to allow people who have similar interests to learn from the authors' work (e.g., https://www.earthsystemcog.org/projects/nuopc/proto_codes/). This would be mostly relevant to the value of this work. While I understand that ICON has a license restriction, it would be useful to present the interface of atmosphere/ocean and their positions in the NUOPC/ESMF layer, without freely releasing the actual code of each model component.

5. The added value of two-way coupling for a high-resolution atmosphere/coastal-ocean model is not clearly demonstrated. Such benefits should be explicitly stated in the conclusion to allow the readers better understand the importance of this work. Some of the figures are redundant, and some of them do not give enough information (see minor points). The authors need to better describe the gains of the coupled simulations for atmosphere and ocean, respectively.

Minor points

1. Section 2.1, Line 60: when mentioning "the usage of nonhydrostatic Euler equations on global domains", I think Gassmann and Herzog (2008) is an important work for ICON and should be cited among others.

2. Lines 65-70: the description here is a little bit disorganized. It would be useful to say something like "The atmospheric component of ICON can be configured to various models (e.g., LES, NWP, climate) by coupling a common dynamical core with different physics packages. The model used in this study is a configuration led by DWD, mainly used for high-resolution NWP applications. Some physics schemes largely inherit the COSMO model."

3. Section 2.1: The YAC library, which is the coupler for ICON-ESM, is also mentioned here. Is it possible for YAC to do the work of this paper?

4. Line 205: pressure levels? It seems to me ICON is using a height-based vertical coordinate.

5. Figures 7 and 8, they are basically telling the same thing as 2-m air temperature is intimately connected with surface temperature.

6. From Fig. 9, it is unclear that the two-way coupled model performs better than the uncoupled one. I understand that after 10 days, temperature is overall enhanced by the coupled model, but such a qualitative comparison is not enough for a scientific journal, especially when demonstrating an issue that is mostly relevant to the value of

this work. I think the authors need some additional quantitative metrics to confirm the improvement (e.g., correlation coefficient, averaged temperature over a certain period).

7. Section 4.1.3, is there any guiding principle to obtain a good load balance in this coupled configuration. How do you draw the current conclusion about the number of cores for ICON and GETM.

References: Gassmann, A., and Herzog, H.-J.: Towards a consistent numerical compressible non-hydrostatic model using generalized Hamiltonian tools, Quarterly Journal of the Royal Meteorological Society, 134, 1597-1613, 10.1002/qj.297, 2008.
* * *

---

## Referee Comment (RC2) · Anonymous Referee #2 · 30 Nov 2020

This paper describes the implementation of the coupling between the atmospheric ICON model and the ocean GETM model using the ESMF/NUOPC coupling technology. It describes in particular the remapping between the unstructured atmosphere grid and the ocean structure grid, and vice-versa, using ESMF exchange grids available in ESMF regridding package. The impact of the two-way coupling is then analysed comparing in detail the results of two simulations of the central Baltic sea, one implementing two-way coupling and the other implementing only one-way coupling from the atmosphere to the ocean. It shows it particular that two-way coupling better

represent the surface temperature as compared to the one-way coupling. The paper is clearly written and easy to follow, and explanations are well illustrated. It represents a nice description of a coupled application and would deserve publication in GMD, but only, I think, if the following major comment is addressed. Major comment: In many places, you write that you implemented conservative interpolation between ICON and GETM, but from what I understood, I think this is not the case because of the non-matching sea-land masks in the two models. Let's take Figure 5 but considering fluxes exchanged from the atmosphere to the ocean. One problem is how to calculate the flux, for example, for the lower left GETM cell. If one normalizes the flux calculation by the whole lower-left cell area ("destarea" option in ESMF and SCRIP), then local conservation is ensured but non-physical values may result; if one normalizes by the intersected area ("fracarea" option in ESMF and SCRIP), then values will be physically sound but local conservation will not be ensured. For example, in Figure 5, it is clear that fluxes coming from the atmosphere in "case-2" regions would be lost as there is no corresponding ocean cell in GETM. The other problem is for the flux coming from case-2 atmosphere region; this part of the flux will not be transferred to any ocean cell and again local conservation will not be ensured. The only way to set up a consistent atmosphere-ocean system and have a well-posed coupled problem, is to adopt the following best practice to defining coherent sea-land masks and sea fractions but it is applicable only if the atmosphere model can consider at least water and land sub surfaces. The original sea-land mask of the ocean model should be taken as is. For the atmosphere model, the fraction of water in each cell should be defined by the conservative remapping of the ocean mask on the atmospheric grid. Therefore, the atmospheric coupling mask should be adapted associating a valid/active index to cells containing at least a fraction of sea. This method ensures that the total sea and land surfaces are the same in the ocean and atmosphere models, allowing global conservation of sea or land integrated quantities. Can you please comment on these important issues and clarify this in your manuscript? Minor comments: • p.1, l.20-21-22: I don't understand why you give the example of the precipitation over sea,

while you start by talking about precipitation over land. I would just remove the "e.g. by precipitation over sea" which is confusing, I think. • p.2, l.43: for the OASIS reference, please use also: Craig A., Valcke S., Coquart L., 2017: Development and performance of a new version of the OASIS coupler, OASIS3-MCT_3.0, Geoscientific Model Development, 10, pp. 3297-3308, doi:10.5194/gmd-10-3297-2017 • p.5, Table 1 captions: You write "If graupel, ice and hail are activated in ICON, then the corresponding contributions to precipitation must also be considered." but these are not explicit in Table 1 right? Maybe you should clarify this. • p.5, Table 1 captions: You write "The humidity quantity is correctly identified by the name of the exchanged ESMF field" but I don't understand what this means. More on this should be provided in the text? • p.5, Table 1 captions: You write "The exchange of flux data (3rd block) or state variables (last block) offers the comparison of different coupling strategies within the same model environment" but I don't understand what this means. More on this should be provided in the text? • p.5, Table 1 captions: The last block is never exchanged as nothing appears in the last column? If so, why does it appear in the Table? • p.13, l.239-240: Can you provide more precise numbers on the load balance obtained with 864 processes for ICON and 384 processes for GETM? • p.14, l.251: can you describe and locate the "upwelling regions" more precisely? • p.13, l.254: It could be relevant to mention Figure 9 when you write about the RV Meteor. • p.13, l.255: It would be helpful to locate the island of Gotland on one figure. • p.13, l.256-258: You state that "the values from the two-way coupled ICONGETM run are in the same range as the measurements and the temporal development also agrees much better with the observations ". I agree this is obviously the case after 10 days but not so obvious for the first days; can you better quantify the improvement, maybe by providing a correlation coefficient. • p. 15, Figure 9: Which area is more precisely concerned, when you write "Easten Gotlan Basin"? Could you give the latitudes and longitudes of the region and maybe show it on one of the figures? • p.15, l.269: Can you locate more precisely the "area east of Oland"? • p.16, l. 274: can you give a definition of "central" and "upper" part of the boundary layer

in meters so to refer to Fig. 12? • p.16, l.277: you write "to the strengthening of the local land-sea circulation (cf. Fig. 11)". I don't clearly see this, can you describe this in more details? • p.17, l.295: You could refer to Figure 15 C and D. • p.18, l.305: What does "cannot be switched off by minor changes" mean? • p.18, l. 310-312: These sentences describe what should be implemented ideally. You should replace "can" by "could" (l.310) and "is done" by "should be done" (l.312) Other comments: • p.1, l.4: replace "The work achieved the development ..." by "We present here the development ..." • p.1, l.19: add "but" before "later" • p.1, l.20-21: Start the sentence with "However, for most ..." and remove it on line 21. • p.2, l.31: Replace "show" by "have" • p.2, l.34-35-36: These sentences use "The latter" and "They" and "them"; I suppose these designate the "coastally trapped waves" but it could be made more explicit for clarity. • p.4, Figure 1 captions: replace "by arrows" with "by horizontal arrows"? • p.4, l.95: consider rewriting the last part of the sentence as "... and only individual specification routines need to be implemented for the model and coupler components." • p.16, l.284: you talk about the surface heat flux, but these are not shown in any figure right? If so, you should add "(not shown)".

Please also note the supplement to this comment:
https://gmd.copernicus.org/preprints/gmd-2020-269/gmd-2020-269-RC2-supplement.pdf

―――――――――――

---

## Author Response (AR1)

Discussion review of anonymous referee #1:

*This manuscript documents the coupling of an unstructured-grid atmospheric model (ICON, configured as a limited-area model) with a structured-grid coastal ocean model (GETM). It clearly describes the technical route and the model simulations. The utilization of a community-based coupler (NUOPC/ESMF) is a good example for other people who have similar interests. I believe that this work fits within the scope of GMD and deserves publication. My major concern is about its scientific quality. While I think GMD appreciates technical work and interdisciplinarity, the current manuscript (as a model description paper) does not offer enough information that could be useful to the general readers. The major conclusion merely summarized what the authors have done: "The demonstration example shows that there is now a coupled model available which allows the investigation of processes at the air-sea interface with high-resolved model simulations." I do believe that the manuscript offers more than that, and it can be further improved. I have some questions and comments which might be helpful to the authors.*

Many thanks to the reviewer for his motivating criticism.

1. *I find one useful aspect of this manuscript is to offer an example of coupling an unstructured-grid atmospheric model with a structured-grid ocean model, based on a community coupler (NUOPC/ESMF). It would be valuable to put the current work into a broader background. Is there any earlier study that has already explored along this line (including global and regional configuration)? If so, the authors should give a general overview; if not, the present work would be more unique and the authors should explicitly speak out.*

   Motivated by this comment, we rewrote the introduction and more clearly present the novelty of our work:

   "There is an ongoing effort to implement the new NUOPC layer into model systems and equip many popular models with a NUOPC interface under the umbrella of the Earth System Prediction Suite [35]. However, until now, there exists only a limited number of publications about its integration. The functioning of the NUOPC layer within the Regional Earth System Model was described by Turuncoglu [36]. Sun et al. [34] developed the regional integrated prediction system SKRIPS based on NUOPC, coupling the atmosphere model WRF and the nonhydrostatic ocean model MITgcm [21]. Only very recently, a coupled unstructured-grid model application consisting of the ocean model ADCIRC [19] and the wave model WAVEWATCH III [38] within the NUOPC-based NOAA Environmental Modeling System (NEMS; `https://www.emc.ncep.noaa.gov/emc/pages/infrastructure/nems.php`) was reported by Moghimi et al. [22].

   Despite the potential of the ESMF echange grid, its implementation and usage in a mediator component has not been published, yet."

2. *What is the major challenge of coupling an unstructured-grid atmospheric model with a structured-grid coastal ocean model? Or more general, any unstructured-grid model (atmosphere/ocean) with a structured-grid model.*

   In Section "3.4 Regridding", we now write:

"One major challenge for the coupling between the unstructured grid of ICON and the structured grid of GETM is the interpolation of data on scattered nodes. The irregularity of the unstructured grid complicates the selection of the stencil. The correct interpolation weights for a conservative interpolation require the determination of the intersections of the source and target grids, and the calculation of the resulting areas. The processing of distributed neighbor information in unstructured grids also requires performant data structures and algorithms. The ESMF exchange grid (`ESMF_XGrid`) and the associated interpolation weights stored in the `ESMF_RouteHandle` hide all these aspects from the user and provide an efficient and automatic conservative interpolation infrastructure."

3. *What is the unique aspect of using NUOPC for this particular work? In comparison with other community-based couplers such as OASIS. It's also useful to briefly review the existing coupled models based on NUOPC.*

We now more clearly state the unique aspects of NUOPC in the introduction:

"Key technical aspects of coupled model systems are the coordinated execution of and the data exchange between the individual models. Required infrastructure for time management, communication between different nodes and interpolation between different grids is provided by various coupling libraries, e.g. MCT, OASIS, ESMF. Coupling frameworks, like the Earth System Modeling Framework [ESMF; 12], provide an additional superstructure layer which offers a standardized execution of models as model components and data exchange in coupler components. On top of ESMF, the National Unified Operational Prediction Capability (NUOPC) layer [35] defines generic components which offer a unified and automated driving of coupled model systems. The generic components require only minimum specialization for the individual models, e.g. registration of routines for initialization and time step advance, definition of required import and possible export data. NUOPC automatically negotiates the data exchange between individual model components based on standard names and synonyms from a dictionary. All required information about the different model grids and their distribution across processors from the models are received during runtime. Therefore, models once equipped with a NUOPC-compliant interface can be plugged into any other coupled model system driven by NUOPC, without the need to adapt coupling specifications.

NUOPC supports a seamless data exchange and interpolation between models operating on different grids via so called connectors. In addition, NUOPC offers mediator components to perform e.g. merging, time-averaging and interface flux calculations on a hub between several models. With ESMF/NUOPC, it is also possible to perform these calculations on automatically generated exchange grids. They have been introduced in Balaji et al. [1] as the union of two rectangular grids. ESMF extended this functionality to unstructured grids, with the final exchange grid obtained by a triangulation of the union. This triangulation is the basis for conservative interpolation. Moreover, the ESMF exchange grid considers the masking of the original grids, e.g. land/sea mask, such that fluxes can

be calculated in a physically consistent way."

For the review of existing coupled models based on NUOPC please see our reply to major point #1 above.

4. *It would be useful to give more details on the construction of a coupled model within NUOPC, for instance, showing some prototype codes to allow people who have similar interests to learn from the authors' work (e.g., https://www.earthsystemcog.org/projects/nuopc/protocodes/). This would be mostly relevant to the value of this work. While I understand that ICON has a license restriction, it would be useful to present the interface of atmosphere/ocean and their positions in the NUOPC/ESMF layer, without freely releasing the actual code of each model component.*

All details on the construction of the coupled model system in the ICON-GETM source code are now freely available from `https://gitlab.com/modellers-tropos/icongetm.git`. The *Code availability* section has been updated with new Zenodo dois for the used ICONGETM version (doi:10.5281/zenodo.4516568 - open access) and for the modified ICON code (doi:10.5281/zenodo.4432739 - restrictive access). In addition, Fig. 1 has been modified, where now all elements of the NUOPC coupling are interactively linked to the corresponding locations in the source code. Furthermore, we now added to Sec. 2.3:

"The implementation of the NUOPC layer in ICONGETM was inspired by the prototype codes `AtmOcnMedPetListProto`, `AtmOcnTransferGridProto`, `CustomFieldDictionaryProto` and `AtmOcnFDSynoProto` as well as `AtmOcnConProto` from `https://earthsystemmodeling.org/nuopc/#prototype-applications`."

5. *The added value of two-way coupling for a high-resolution atmosphere/coastal ocean model is not clearly demonstrated. Such benefits should be explicitly stated in the conclusion to allow the readers better understand the importance of this work. Some of the figures are redundant, and some of them do not give enough information (see minor points). The authors need to better describe the gains of the coupled simulations for atmosphere and ocean, respectively.*

The well-known added value of two-way coupling for high-resolution atmosphere-ocean models is now stated in the introduction and used to clearly motivate our work with focus on the technical implementation of the latest ESMF/NUOPC coupling technologies:

"In numerous studies, the added value of two-way coupled atmosphere-ocean models has been demonstrated. Interactive model coupling is important for representing the mutual interactions and feedbacks between atmosphere and ocean dynamics [e.g., 6]. The sea surface temperature (SST) of the ocean determines moisture fluxes into the atmosphere and the stability of the atmospheric boundary layer [10]. The modulated surface wind in turn affects surface currents and mixing in the ocean, both altering SST patterns. This air-sea interaction is very dynamic and strongly sensitive to fronts and eddies [33, 29]. In the coastal ocean, fronts are further pronounced due to upwelling and river run-off. Therefore, especially high-resolution coastal applications, where sharp gradients and small-scale eddies are resolved, can benefit from two-way coupled atmosphere-ocean

models.

The atmosphere model COAMPS [13] and the regional ocean model ROMS [30] were coupled with the Model Coupling Toolkit [MCT; 15] for investigating an upwelling event with a $1\,km$ high resolution [23]. In the following decade, numerous high resolution studies were performed with the two-way coupled model system COAMPS-NCOM, in which COAMPS was originally coupled via MCT with the coastal ocean model NCOM [2]. Pullen et al. [26, 24] demonstrated the improved skill of the two-way coupled model system during Bora events in the Adriatic Sea, simulated down to a resolution of $4\,km$ in the atmosphere and $2\,km$ in the ocean. With the same resolution and a coupling time step of $12\,min$, the model system has been applied to the Ligurian Sea and confirmed the importance of the interactive model coupling in the coastal zone [32]. The impact of coastal orography was investigated in a $2\,km$ simulation of Madeira Island [25]. Another two-way coupled model system widely applied in high-resolution studies is COAWST [37]. The atmosphere model WRF [31], ROMS and the wave model SWAN [3] are coupled with MCT. COAWST has been applied for a realistic hindcast of a storm event over the Gulf of Lion and the Balearic Seas with a resolution of $3\,km$ in the atmosphere and $1.8\,km$ in the ocean [28]. In another application, a Bora event and the dense water formation in the Adriatic sea with $7\,km$ resolution in the atmosphere and $1\,km$ in the ocean was simulated [5]. Both studies investigated the effects of different coupling strategies and demonstrated the benefit of the fully coupled model system. Recently, the high-resolution regional coupled environmental prediction system UKC for the northwest European Shelf has been developed [17, 18]. On a $1.5\,km$ high resolution, the atmosphere model MetUM [9, 4] was coupled with the ocean model NEMO [20] via OASIS3-MCT [8]. First results demonstrate reduced bias in SST fields [16] and impacts on cloud and fog formation over the North Sea [11, 10]."

We refer to the effects of the two-way coupling in Sec. 4.2, when we evaluate the results of our demonstration example for the successfully developed coupled model system.

However, an in-depth analysis of the high-resolution air-sea interactions during a specific event and focused on a local scale is out of the scope for our initial technical model description paper and planned as a follow-up study. Instead, the focus of our paper is the added value and potential of using the ESMF exchange grid in ICONGETM, which is now discussed in more detail in the Discussion section:

"ICONGETM supports the exchange of fluxes and state variables across the air-sea interface. The applied ESMF exchange grid guarantees a conservative flux exchange. The NUOPC-Mediator performs additional unit conversion and merging of precipitation fluxes, see Tab. 1. In ICONGETM v1.0, the air-sea fluxes are taken from the atmosphere model ICON. Their calculation in ICON is very complex and deeply nested in the model code. However, in later releases the air-sea fluxes should be calculated in the mediator, in terms of state variables received from atmosphere and ocean. Their calculation directly on the ESMF exchange grid also solves the problem of different land/sea masks [1] and ensures physical consistency in the sense that no fluxes calculated over land, i.e. not influenced by the sea surface temperature, are provided to the ocean. Without an ESMF exchange grid creep, nearest neighbor and other extrapolation methods might be required [see e.g. 14, 7, 36], especially if an atmosphere model with low spatial resolution is coupled. Fluxes provided by the mediator can be applied in the atmosphere and ocean over the same period until new fluxes are calculated in the next coupling time step. Besides this physical and energetic consistency, the flux calculation on the ESMF exchange grid in a central mediator component also offers the most straight-forward extension of the coupled system by models for e.g. waves and sea ice."

This is now also stated in the conclusion.

Regarding the redundancy of the figures please see our reply to minor point #5 below.

Minor points:

1. *Section 2.1, Line 60: when mentioning "the usage of nonhydrostatic Euler equations on global domains", I think Gassmann and Herzog (2008) is an important work for ICON and should be cited among others.*

    Added reference.

2. *Lines 65-70: the description here is a little bit disorganized. It would be useful to say something like "The atmospheric component of ICON can be configured to various models (e.g., LES, NWP, climate) by coupling a common dynamical core with different physics packages. The model used in this study is a configuration led by DWD, mainly used for high-resolution NWP applications. Some physics schemes largely inherit the COSMO model."*

    Added and adopted the suggested details to the description.

3. *Section 2.1: The YAC library, which is the coupler for ICON-ESM, is also mentioned here. Is it possible for YAC to do the work of this paper?*

    Sure, but the implementational effort would be high. According to the latest documentation from `https://dkrz-sw.gitlab-pages.dkrz.de/yac/`, YAC does not offer the same built-in functionality as NUOPC e.g. generic automated driving of coupled model systems and the features of the ESMF exchange grid.

4. *Line 205: pressure levels? It seems to me ICON is using a height-based vertical coordinate.*

    It is now clearly stated:

    "The vertical terrain-following hybrid grids consist of 90, 65 and 54 height-based vertical levels. The heights are pre-defined depending on the associated pressure in the US 1976 standard atmosphere, with the top boundary of the model domain depending on the numbers of levels [27, Fig. 3.5]."

5. *Figures 7 and 8, they are basically telling the same thing as 2-m air temperature is intimately connected with surface temperature.*

    Agreed, (sea) surface and $2\,m$ air temperature are closely related. On the other hand, while the sea surface temperature in either case only

represents a lower boundary condition for the atmospheric model, the $2\,m$ air temperature actually shows a response of the ICON model, which was important for us to show. This is also reflected for example by signatures of ocean eddies in SST as well as effect of land and uncoupled ocean surface on $2\,m$ air temperature.

6. *From Fig. 9, it is unclear that the two-way coupled model performs better than the uncoupled one. I understand that after 10 days, temperature is overall enhanced by the coupled model, but such a qualitative comparison is not enough for a scientific journal, especially when demonstrating an issue that is mostly relevant to the value of this work. I think the authors need some additional quantitative metrics to confirm the improvement (e.g., correlation coefficient, averaged temperature over a certain period).*

   A short statistical evaluation is now added to Sec. 4.2.1:

   "The average deviation from the modelled and measured temperature is about $1.6\,K$ / $1.5\,K$ and $1.9\,K$ / $2.0\,K$ for the two-way coupled and uncoupled simulations from 01 / 10 July 2012 onward, respectively. This is a significant improvement of about 15%/ 25%, respectively. However, the Pearson correlation coefficient is only slightly improved, i.e. 0.7158 / 0.7487 and 0.6996 / 0.7336 for the two-way coupled and uncoupled simulations from 01 / 10 July 2012, respectively. The more reduced average deviation and higher correlation of the two-way coupled simulations after 10 July 2012 is related to the spin up of the model, since GETM is initialized as hot start while ICON uses the IFS reanalysis data."

7. *Section 4.1.3, is there any guiding principle to obtain a good load balance in this coupled configuration. How do you draw the current conclusion about the number of cores for ICON and GETM.*

   The optimal load-balancing was estimated empirically in terms of minimum idle/waiting times for the single model components. The sentence in the text has been modified:

   "For the present setup a good concurrent load-balancing **with minimum idle/waiting times for the single model components was empirically** obtained with 864 processes for ICON and 384 processes for GETM."

Discussion review of anonymous referee #2:

*This paper describes the implementation of the coupling between the atmospheric ICON model and the ocean GETM model using the ESMF/NUOPC coupling technology. It describes in particular the remapping between the unstructured atmosphere grid and the ocean structure grid, and vice-versa, using ESMF exchange grids available in ESMF regridding package. The impact of the two-way coupling is then analysed comparing in detail the results of two simulations of the central Baltic sea, one implementing two-way coupling and the other implementing only one-way coupling from the atmosphere to the ocean. It shows it particular that two-way coupling better represent the surface temperature as compared to the one-way coupling. The paper is clearly written and easy to follow, and explanations are well illustrated. It represents a nice description of a coupled application and would deserve publication in GMD, but only, I think, if the following major comment is addressed.*

Many thanks to the Reviewer for his motivating critism.

Major comment:

*In many places, you write that you implemented conservative interpolation between ICON and GETM, but from what I understood, I think this is not the case because of the non-matching sea-land masks in the two models. Letâ ĂŽs take Figure 5 but considering fluxes exchanged from the atmosphere to the ocean. One problem is how to calculate the flux, for example, for the lower left GETM cell. If one normalizes the flux calculation by the whole lower-left cell area ("destarea" option in ESMF and SCRIP), then local conservation is ensured but non-physical values may result; if one normalizes by the intersected area ("fracarea" option in ESMF and SCRIP), then values will be physically sound but local conservation will not be ensured.*

In ICONGETM, the interpolation is carried out via the ESMF exchange grid. This two-step procedure from the source to the exchange grid and further to the destination grid is a combination of the mentioned individual interpolation methods for a direct interpolation from a source to a destination grid. Therefore, interpolation via the ESMF exchange grid guarantees global conservation and physically reasonable interpolated quantities.

*For example, in Figure 5, it is clear that fluxes coming from the atmosphere in "case-2" regions would be lost as there is no corresponding ocean cell in GETM. The other problem is for the flux coming from case-2 atmosphere region; this part of the flux will not be transferred to any ocean cell and again local conservation will not be ensured.*

A conservative interpolation ensures that e.g. the energy exchanged through a **common** area is conserved. For the data exchange between the atmosphere and ocean in ICONGETM, this is guaranteed by the implementation and use of the ESMF exchange grid. Of course, fluxes leaving the atmosphere not towards a common area with GETM are not further accounted in the atmosphere-ocean-system. A conservative atmosphere-ocean-system requires the surface area of the sea water fraction in an ICON cell being identical to the corresponding area in the exchange grid with GETM, see also next point.

*The only way to set up a consistent atmosphere-ocean system and have a well-posed coupled problem, is to adopt the following best practice to defining coherent sea-land masks and sea fractions but it is applicable only if the atmosphere model can consider at least water and land sub surfaces. The original sea-land mask of the ocean model should be taken as is. For the atmosphere model, the fraction of*

*water in each cell should be defined by the conservative remapping of the ocean mask on the atmospheric grid. Therefore, the atmospheric coupling mask should be adapted associating a valid/active index to cells containing at least a fraction of sea. This method ensures that the total sea and land surfaces are the same in the ocean and atmosphere models, allowing global conservation of sea or land integrated quantities. Can you please comment on these important issues and clarify this in your manuscript?*

We absolutely agree with the reviewer. We double-checked the ICON code whether it is possible to implement this treatment, but modifications are far from trivial, at least for us, who are no developers of the ICON core. In the new Discussion section we now write: "Another feature missing in ICON is mixed land/ocean cells. However, for a fully coherent treatment of land/sea masks in the coupled system, ICON needs to consider the water fraction area of GETM from the exchange grid."

Minor comments:

- *p.1, l.20-21-22: I don't understand why you give the example of the precipitation over sea, while you start by talking about precipitation over land. I would just remove the "e.g. by precipitation over sea" which is confusing, I think.*

  You are absolutely right. This sentence is now removed from the introduction.

  (After reorganizing the introduction, this part is now removed.)

- *p.2, l.43: for the OASIS reference, please use also: Craig A., Valcke S., Coquart L., 2017: Development and performance of a new version of the OASIS coupler, OASIS3-MCT_3.0, Geoscientific Model Development, 10, pp. 3297-3308, doi:10.5194/gmd-10-3297-2017*

  Added reference.

- *p.5, Table 1 captions: You write "If graupel, ice and hail are activated in ICON, then the corresponding contributions to precipitation must also be considered." but these are not explicit in Table 1 right? Maybe you should clarify this.*

  The sentence has been rephrased for clarification: "The corresponding contributions to precipitation from graupel, hail and ice are only considered for the coupling if they are activated in ICON."

  Graupel, ice and hail have been added to the table.

- *p.5, Table 1 captions: You write "The humidity quantity is correctly identified by the name of the exchanged ESMF field" but I don't understand what this means. More on this should be provided in the text?*

  It is now clearly written:

  "The **exchanged** humidity quantity (**dew point or relative humidity**) is correctly identified by the name attribute of the connected ESMF field".

- *p.5, Table 1 captions: You write "The exchange of flux data (3rd block) or state variables (last block) offers the comparison of different coupling strategies within the same model environment" but I don't understand what this means. More on this should be provided in the text?*

Modified sentence:

"**The possibility to exchange either** flux data (3rd block) **or** state variables (last block) offers the comparison of different coupling strategies within the same model environment."

- *p.5, Table 1 captions: The last block is never exchanged as nothing appears in the last column? If so, why does it appear in the Table?*

Because in Tab. 1 all quantities are listed that can be exchanged with the developed model system, not only the ones considered in the demonstration example.

- *p.13, l.239-240: Can you provide more precise numbers on the load balance obtained with 864 processes for ICON and 384 processes for GETM?*

The optimal load-balancing was estimated empirically in terms of minimum idle/waiting times for the single model components. A systematic analysis was not conducted, because it would be applicable only for this specific setup anyway.

- *p.14, l.251: can you describe and locate the "upwelling regions" more precisely?*

The text was expanded accordingly:

"In July 2012, the simulated SST ranged around $289\,K$, with values below $282\,K$ in the upwelling areas south of the coast of mainland Sweden and the islands of Öland and Gotland."

- *p.13, l.254: It could be relevant to mention Figure 9 when you write about the RV Meteor.*

Added figure reference to text.

- *p.13, l.255: It would be helpful to locate the island of Gotland on one figure.*

The white frames in Figs. 2 and 9 are showing the island of Gotland.

- *p.13, l.256-258: You state that "the values from the two-way coupled ICONGETM run are in the same range as the measurements and the temporal development also agrees much better with the observations". I agree this is obviously the case after 10 days but not so obvious for the first days; can you better quantify the improvement, maybe by providing a correlation coefficient.*

A short statistical evaluation is now added to Sec. 4.2.1:

"The average deviation from the modelled and measured temperature is about $1.6\,K$ / $1.5\,K$ and $1.9\,K$ / $2.0\,K$ for the two-way coupled and uncoupled simulations from 01 / 10 July 2012 onward, respectively. This is a significant improvement of about 15%/ 25%, respectively. However, the Pearson correlation coefficient is only slightly improved, i.e. 0.7158 / 0.7487 and 0.6996 / 0.7336 for the two-way coupled and uncoupled simulations from 01 / 10 July 2012, respectively. The more reduced average deviation and higher correlation of the two-way coupled simulations after 10 July 2012 is related to the spin up of the model, since GETM is initialized as hot start while ICON uses the IFS reanalysis data."

- *p. 15, Figure 9: Which area is more precisely concerned, when you write "Eastern Gotland Basin"? Could you give the latitudes and longitudes of the region and maybe show it on one of the figures?*

  The ship track is now presented on the right panel of Fig. 9.

- *p.15, l.269: Can you locate more precisely the "area east of Öland"?*

  Added reference to marker in Fig. 8.

- *p.16, l. 274: can you give a definition of "central" and "upper" part of the boundary layer in meters so to refer to Fig. 12?*

  It is now clearly written: "... in the central to upper part of the boundary layer, between $900\,m$ and $2400\,m$ in the left panel of Fig. 12. Due to reduced evaporation, it is less in the lowermost part, below $500\,m$ in the left panel of Fig. 12."

- *p.16, l.277: you write "to the strengthening of the local land-sea circulation (cf. Fig. 11)". I don't clearly see this, can you describe this in more details?*

  The description has been reworded:

  "In addition, there is less momentum mixed downwards (not shown), which is a likely explanation for the locally reduced wind velocity in the upwelling regions at Sweden's mainland coast and the Öland and Gotland islands, shown by negative differences in the central panel of Fig. 11. In the coupled case, the temperature gradient between land and sea is increased in the area of the upwelling, cf. Fig. 8, with almost the same land temperatures but significantly lower SSTs, which locally increases the onshore wind component and thus weakens the overall more easterly wind in Fig. 11."

- *p.17, l.295: You could refer to Figure 15 C and D.*

  Done as suggested.

- *p.18, l.305: What does "cannot be switched off by minor changes" mean?*

  We now write:

  "In ICONGETM v1.0, the air-sea fluxes are taken from the atmosphere model ICON. Their calculation in ICON is very complex and deeply nested in the model code."

  Therefore, the flux calculation cannot easily be moved to the mediator.

- *p.18, l. 310-312: These sentences describe what should be implemented ideally. You should replace "can" by "could" (l.310) and "is done" by "should be done" (l.312)*

  The whole part of the discussion on flux exchange via a mediator has been modified.

Other comments:

- *p.1, l.4: replace "The work achieved the development ..." by "We present here the development ..."*

  Modified sentence.

- *p.1, l.19: add "but" before "later"*

  After reorganizing the introduction, this part is now removed.

- *p.1, l.20-21: Start the sentence with "However, for most ..." and remove it on line 21.*

  After reorganizing the introduction, this part is now removed.

- *p.2, l.31: Replace "show" by "have"*

  After reorganizing the introduction, this part is now removed.

- *p.2, l.34-35-36: These sentences use "The latter" and "They" and "them"; I suppose these designate the "coastally trapped waves" but it could be made more explicit for clarity.*

  After reorganizing the introduction, this part is now removed.

- *p.4, Figure 1 captions: replace "by arrows" with "by horizontal arrows"?*

  No, all arrows repesent generic NUOPC operations.

- *p.4, l.95: consider rewriting the last part of the sentence as "... and only individual specification routines need to be implemented for the model and coupler components."*

  Rephrased sentence.

- *p.16, l.284: you talk about the surface heat flux, but these are not shown in any figure right? If so, you should add "(not shown)".*

  Added "not shown".

**Bibliography**

[1] V. Balaji et al. "The Exchange Grid: A mechanism for data exchange between Earth System components on independent grids". In: *Parallel Computational Fluid Dynamics 2005*. Elsevier, 2006, pp. 179–186. DOI: 10.1016/B978-044452206-1/50021-5.

[2] Charlie N. Barron et al. "Formulation, implementation and examination of vertical coordinate choices in the Global Navy Coastal Ocean Model (NCOM)". In: *Ocean Modelling* 11.3-4 (2006), pp. 347–375. ISSN: 14635003. DOI: 10.1016/j.ocemod.2005.01.004.

[3] N. Booij, R. C. Ris, and L. H. Holthuijsen. "A third-generation wave model for coastal regions: 1. Model description and validation". In: *Journal of Geophysical Research: Oceans* 104.C4 (1999), pp. 7649–7666. ISSN: 01480227. DOI: 10.1029/98JC02622.

[4] Andrew Brown et al. "Unified Modeling and Prediction of Weather and Climate: A 25-Year Journey". In: *Bulletin of the American Meteorological Society* 93.12 (2012), pp. 1865–1877. ISSN: 1520-0477. DOI: 10.1175/BAMS-D-12-00018.1.

[5] Sandro Carniel et al. "Scratching beneath the surface while coupling atmosphere, ocean and waves: Analysis of a dense water formation event". In: *Ocean Modelling* 101 (2016), pp. 101–112. ISSN: 14635003. DOI: 10.1016/j.ocemod.2016.03.007.

[6] Dudley B. Chelton and Shang-Ping Xie. "Coupled Ocean-Atmosphere Interaction at Oceanic Mesoscales". In: *Oceanography* 23.4 (2010), pp. 52–69. ISSN: 10428275. DOI: 10.5670/oceanog.2010.05.

[7] Sue Chen et al. "Effect of Two-Way Air–Sea Coupling in High and Low Wind Speed Regimes". In: *Monthly Weather Review* 138.9 (2010), pp. 3579–3602. ISSN: 1520-0493. DOI: 10.1175/2009MWR3119.1.

[8] Anthony Craig, Sophie Valcke, and Laure Coquart. "Development and performance of a new version of the OASIS coupler, OASIS3-MCT_3.0". In: *Geoscientific Model Development* 10.9 (2017), pp. 3297–3308. ISSN: 1991-9603. DOI: 10.5194/gmd-10-3297-2017.

[9] Michael J.P. Cullen. "The unified forecast/climate model". In: *Meteorological Magazine* 122 (1993), pp. 81–94.

[10] Joachim Fallmann et al. "Impact of high-resolution ocean–atmosphere coupling on fog formation over the North Sea". In: *Quarterly Journal of the Royal Meteorological Society* 145.720 (2019), pp. 1180–1201. ISSN: 0035-9009. DOI: 10.1002/qj.3488.

[11] Joachim Fallmann et al. "Impact of sea surface temperature on stratiform cloud formation over the North Sea". In: *Geophysical Research Letters* 44.9 (2017), pp. 4296–4303. ISSN: 00948276. DOI: 10.1002/2017GL073105.

[12] Chris Hill et al. "The architecture of the earth system modeling framework". In: *Computing in Science & Engineering* 6.1 (2004), pp. 18–28. ISSN: 1521-9615. DOI: 10.1109/MCISE.2004.1255817.

[13] Richard M. Hodur. "The Naval Research Laboratory's Coupled Ocean/Atmosphere Mesoscale Prediction System (COAMPS)". In: *Monthly Weather Review* 125.7 (1997), pp. 1414–1430. ISSN: 0027-0644. DOI: 10.1175/1520-0493(1997) 125<1414:TNRLSC>2.0.CO;2.

[14] A. Birol Kara, Alan J. Wallcraft, and Harley E. Hurlburt. "A Correction for Land Contamination of Atmospheric Variables near Landâ ĂŞSea Boundaries*". In: *Journal of Physical Oceanography* 37.4 (2007), pp. 803–818. ISSN: 1520-0485. DOI: 10.1175/JPO2984.1.

[15] Jay Larson, Robert Jacob, and Everest Ong. "The Model Coupling Toolkit: A New Fortran90 Toolkit for Building Multiphysics Parallel Coupled Models". In: *The International Journal of High Performance Computing Applications* 19.3 (2005), pp. 277–292. ISSN: 1094-3420. DOI: 10.1177/1094342005056115.

[16] Huw W. Lewis et al. "Evaluating the impact of atmospheric forcing and airâ ĂŞsea coupling on near-coastal regional ocean prediction". In: *Ocean Science* 15.3 (2019), pp. 761–778. ISSN: 1812-0792. DOI: 10.5194/os-15-761-2019.

[17] Huw W. Lewis et al. "The UKC2 regional coupled environmental prediction system". In: *Geoscientific Model Development* 11.1 (2018), pp. 1–42. ISSN: 1991-9603. DOI: 10.5194/gmd-11-1-2018.

[18] Huw W. Lewis et al. "The UKC3 regional coupled environmental prediction system". In: *Geoscientific Model Development* 12.6 (2019), pp. 2357–2400. ISSN: 1991-9603. DOI: 10.5194/gmd-12-2357-2019.

[19] R. A. Luettich, Jr., J. J. Westerink, and Norman W. Scheffner. *ADCIRC: An Advanced Three-Dimensional Circulation Model for Shelves, Coasts, and Estuaries. Report 1. Theory and Methodology of ADCIRC-2DDI and ADCIRC-3DL*. Tech. rep. Vicksburg, MS: Coastal Engineering Research Center, 1992.

[20] Gurvan Madec et al. *NEMO ocean engine (Version v3.6-patch)*. Tech. rep. Pôle De Modélisation De L'institut Pierre-simon Laplace (IPSL), 2017. DOI: 10.5281/ZENODO.3248739.

[21] John Marshall et al. "A finite-volume, incompressible Navier Stokes model for studies of the ocean on parallel computers". In: *Journal of Geophysical Research: Oceans* 102.C3 (1997), pp. 5753–5766. ISSN: 01480227. DOI: 10.1029/96JC02775.

[22] Saeed Moghimi et al. "Development of an ESMF Based Flexible Coupling Application of ADCIRC and WAVEWATCH III for High Fidelity Coastal Inundation Studies". In: *Journal of Marine Science and Engineering* 8.5 (2020), p. 308. ISSN: 2077-1312. DOI: 10.3390/jmse8050308.

[23] Natalie Perlin et al. "Numerical Simulation of Airâ ĂŞSea Coupling during Coastal Upwelling". In: *Journal of Physical Oceanography* 37.8 (2007), pp. 2081–2093. ISSN: 1520-0485. DOI: 10.1175/JPO3104.1.

[24] Julie Pullen et al. "Bora event variability and the role of air-sea feedback". In: *Journal of Geophysical Research* 112.C3 (2007), C03S18. ISSN: 0148-0227. DOI: 10.1029/2006JC003726.

[25] Julie Pullen et al. "Modeling the air‐sea feedback system of Madeira Island". In: *Journal of Advances in Modeling Earth Systems* 9.3 (2017), pp. 1641–1664. ISSN: 1942-2466. DOI: 10.1002/2016MS000861.

[26] Julie Pullen, James D. Doyle, and Richard P. Signell. "Two-Way Air–Sea Coupling: A Study of the Adriatic". In: *Monthly Weather Review* 134.5 (2006), pp. 1465–1483. ISSN: 1520-0493. DOI: 10.1175/MWR3137.1.

[27] Daniel Reinert et al. *Database Reference Manual for ICON and ICON-EPS*. Tech. rep. Offenbach: DWD, 2020. DOI: 10.5676/DWD_pub/nwv/icon_1.2.12.

[28] Lionel Renault et al. "Coupled atmosphere-ocean-wave simulations of a storm event over the Gulf of Lion and Balearic Sea". In: *Journal of Geophysical Research: Oceans* 117.C9 (2012), n/a–n/a. ISSN: 01480227. DOI: 10.1029/2012JC007924.

[29] Mingming Shao et al. "The Variability of Winds and Fluxes Observed Near Submesoscale Fronts". In: *Journal of Geophysical Research: Oceans* 124.11 (2019), pp. 7756–7780. ISSN: 2169-9275. DOI: 10.1029/2019JC015236.

[30] Alexander F. Shchepetkin and James C. McWilliams. "The regional oceanic modeling system (ROMS): a split-explicit, free-surface, topography-following-coordinate oceanic model". In: *Ocean Modelling* 9.4 (2005), pp. 347–404. ISSN: 14635003. DOI: 10.1016/j.ocemod.2004.08.002.

[31] William C. Skamarock et al. *A Description of the Advanced Research WRF Version 2*. Tech. rep. UCAR, 2005. DOI: 10.5065/D6DZ069T.

[32] R. J. Small et al. "Air–Sea Interaction in the Ligurian Sea: Assessment of a Coupled Ocean–Atmosphere Model Using In Situ Data from LASIE07". In: *Monthly Weather Review* 139.6 (2011), pp. 1785–1808. ISSN: 0027-0644. DOI: 10.1175/2010MWR3431.1.

[33] R. J. Small et al. "Air–sea interaction over ocean fronts and eddies". In: *Dynamics of Atmospheres and Oceans* 45.3-4 (2008), pp. 274–319. ISSN: 03770265. DOI: 10.1016/j.dynatmoce.2008.01.001.

[34] Rui Sun et al. "SKRIPS v1.0: a regional coupled ocean–atmosphere modeling framework (MITgcm–WRF) using ESMF/NUOPC, description and preliminary results for the Red Sea". In: *Geoscientific Model Development* 12.10 (2019), pp. 4221–4244. ISSN: 1991-9603. DOI: 10.5194/gmd-12-4221-2019.

[35] Gerhard Theurich et al. "The earth system prediction suite: Toward a coordinated U.S. modeling capability". In: *Bulletin of the American Meteorological Society* 97.7 (2016), pp. 1229–1247. ISSN: 00030007. DOI: 10.1175/BAMS-D-14-00164.1.

[36] Ufuk Utku Turuncoglu. "Toward modular in situ visualization in Earth system models: the regional modeling system RegESM 1.1". In: *Geoscientific Model Development* 12.1 (2019), pp. 233–259. ISSN: 1991-9603. DOI: 10.5194/gmd-12-233-2019.

[37]  John C. Warner et al. "Development of a Coupled Ocean–Atmosphere–Wave–Sediment Transport (COAWST) Modeling System". In: *Ocean Modelling* 35.3 (2010), pp. 230–244. ISSN: 14635003. DOI: 10.1016/j.ocemod.2010.07.010.

[38]  The WAVEWATCH III Development Group WW3DG. *User Manual and System Documentation of WAVEWATCH III version 6.07.* Tech. rep. Tech. Note 33. NOAA/NWS/NCEP/MMAB, 2019.

---

## Referee Report (RR1)

2nd review of "ICONGETM v1.0 – Flexible NUOPC-driven two-way coupling via ESMF exchange grids between the unstructured-grid atmosphere model ICON and the structured-grid coastal ocean model GETM"

Major comment

I consider that the authors have carefully analyzed the remarks I formulated for the first review but I still think that my main remark about the impossibility to have a fully conservative interpolation with non-matching sea-land mask between the ocean and the atmosphere is not answered properly. Contrary to what the authors state in their reply (" Therefore, interpolation via the ESMF exchange grid guarantees global conservation …"), the exchange grid ensures **locally** conservative data exchange but cannot ensure the global conservation if the sea-land masks of the two models do not match, and this should be clarified in the text. Furthermore, the authors seem to agree with my analusis when they write "A conservative atmosphere-ocean-system requires the surface area of the sea water fraction in an ICON cell being identical to the corresponding area in the exchange grid, see also next point." and when they discuss the possibility/difficulty to implement mixed land/ocean cells in ICON. Therefore, I strongly suggest that the following sentences be modified so not to mislead the reader:

- L.4 : add "locally" before "conservative data exchange via ESMF exchange grids"
- L.54-55: at the end of the sentence, add: ", even if the exchange grid cannot force their global conservation if the sea-land masks do not match between the ocean and the atmosphere models.".
- L.220: At the end of this paragraph: "It is also obvious that in case 2, a part of the flux calculated by the atmospheric cells will be lost as it cannot be attributed to any ocean cell in GETM; the global conservation of the fluxes cannot be ensured  if the sea-land masks do not match between the ocean and the atmosphere models."
- L.351-352: Modify the sentence for "The applied ESMF exchange grid guarantees a conservative flux exchange, except in the case of non-matching sea-land masks between the ocean and the atmosphere."
- L.355-356: Modify the sentence for "Their calculation directly on the ESMF exchange, even if it cannot solve the problem of different land-sea masks (Balaji et al., 2006) ensures physical consistency …"
- L.365-366: Modify the sentence "Their calculation directly on the ESMF exchange grid also solves the problem of different land/sea masks (Balaji et al., 2006) and ensures physical consistency in the sense that no fluxes calculated over land, i.e. not influenced by the sea surface temperature, are provided to the ocean." by something like :

  "Even if the ESMF exchange grid does not solve the problem of different land/sea masks (Balaji et al., 2006), it ensures physical consistency in the sense that no fluxes calculated over land, i.e. not influenced by the sea surface temperature, are provided to the ocean.

  Regarding the problem of matching land/sea masks between the atmosphere and the ocean, it is worth mentioning here that the only way to have a well-posed coupled problem, is to adopt the following best practice, which is applicable only if the atmosphere model can consider water and land sub surfaces. The original sea-land mask of the ocean model should be taken as is. For the atmosphere model, the fraction of water in each cell should be defined by the conservative remapping of the ocean mask on the atmospheric grid. Then, the atmospheric coupling mask should be adapted associating a valid/active index to cells containing at least a fraction of sea. This method ensures that the total sea and land surfaces are the same in the ocean and atmosphere models, allowing global conservation of sea or land integrated quantities. ICON mask was not defined following this best practice (and it would involve some non-trivial modifications to do so, so the global conservation of fluxes cannot be fully ensured in the current coupled model."

Other important comments:

- p.14, l.272: I don't understand what "a good concurrent load-balancing with minimum idle/waiting times for the single model components was empirically …" means. Do you mean that the elapsed time for running ICON as single model on 864 processes was almost the same as the

elapsed time for running GETM as single model on 384 processes, and therefore you suppose that using these number of processes for each component in the coupled system will lead to minimum idle/waiting time? If so, it should be rephrased for something like "For the present set-up, ICON was run on 864 processes and GETM on 384 processes. It is supposed that this distribution leads to minimum idle/waiting time of any of the component as the elapsed time for running ICON as a single model on 864 processes was about the same than the elapsed time for running GETM as a single model on 384 processes."

- p.15, l.292-293: The statistics presented are extremely difficult to understand. I suppose that 1.6 K/1.5K are for the two-way coupled simulation and that 1.9K/2.0K are for the uncoupled simulation. But e.g. for the two-way coupled, I don't understand what the two numbers (1.6 K and 1.5K) relate to; are these for different averaging periods (maybe 01-10 July and 10 July onward?) ? What does "01/10 July 2012 onward" stand for? I have the same remark for the Pearson coefficient. This remark about the need to better quantify the improvement brought by the two-way coupling was done by myself and by the other reviewer. I consider that the answer brought by the author is not satisfying, at least under the current form.

Minor comments

- p.2, l.26: I suppose that the sentence "The atmosphere model WRF …with MCT." describes COAWST? If so, it would be clearer by linking the two sentences with something like: "is COAWST (Warner et al., 2010) into which the atmosphere model WRF … with MCT."
- p.2, l.27: "the ocean model" is missing before "ROMS"
- p.2, l.37 : consider changing "… are the coordinated execution of and the data exchange between the individual models." with "… are the coordinated execution of the individual component models and the data exchange between these models."
- p.2, l.39: It looks like you are doing a distinction between "coupling libraries" and "coupling frameworks" which is fine to me. But ESMF is mentioned as an example of both categories. In the coupling library list, you should replace OASIS by OASIS3-MCT (which has been introduced just above), you should remove "ESMF" and maybe replace it with "YAC (Yet Another Coupler, Hanke et al., 2016)" and put the reference to YAC there in the text (and remove it at l.97).
- p.2, l.47: consider changing "from the models are received during runtime" for "are received during runtime from the models"
- p.2, l.52: I don't think that Balaji restricted his definition of an exchange grid to two rectangular grids. Therefore, please consider changing the two sentences for "They have been introduced in Balaji et al. (2006). ESMF implements this functionality to unstructured grids, …"
- p.3, l.66, and p.22, l.372: why do you call ICON a "next-generation" atmosphere; ICON exists today so it is not a next-generation" model; please consider changing "next-generation" for "state-of-the-art" or something similar.
- p.3, l.85: I think a verb is missing in "can be configured to various models", maybe "can be configured to produce various models"
- p.4, l.100: the link under www.getm.eu does not work (at least for me). Should it be "https://getm.eu" ?
- p.6, Table 1 captions: consider adding "although exchange of state variables is not activated in the simulations reported in this paper" after "same model environment."
- p.7 l.133: consider adding ", although exchange of state variables is not activated in the simulations reported in this paper" after "the exchange of flux data". Make a new sentence for "See Tab. 1 for a list …"
- p.7, l.139: replace "will be used" by "are used".
- p.7, l.142: add "the" before "Initialization phase".
- p.7, l.159: why do you write "compare with Fig.1" and not simply "see Fig.1"?
- p.8, l.182: replace "domain distributing" by "domain distribution"
- p.13, l.234: add a coma after "Baltic Sea setup"
- p.14, l.272: add a coma after "For the present setup"
- p.20, l.346: add "see" before "Fig. 15 D"
- p.20, l.347, add "see" before "Fig. 15 A and C"

---

## Author Response (AR2)

Discussion 2nd review of Sophie Valcke:

2nd review of "ICONGETM v1.0 – Flexible NUOPC-driven two-way coupling via ESMF exchange grids between the unstructured-grid atmosphere model ICON and the structured-grid coastal ocean model GETM"

Major comment:

*I consider that the authors have carefully analysed the remarks I formulated for the first review but I still think that my main remark about the impossibility to have a fully conservative interpolation with non-matching sea-land mask between the ocean and the atmosphere is not answered properly. Contrary to what the authors state in their reply (" Therefore, interpolation via the ESMF exchange grid guarantees global conservation ..."), the exchange grid ensures locally conservative data exchange but cannot ensure the global conservation if the sea-land masks of the two models do not match, and this should be clarified in the text.*

We are very grateful for pointing again to the still unclear explanation in our manuscript. In terms of conservativity, we would like to stress the difference between *interpolation* and *flux exchange*, which we now clearly explain in the manuscript. The interpolation is only defined and performed over the area of the exchange grid. Therefore, *the interpolation is always conservative by definition.* However, in case the atmosphere and ocean models do not share the same common sea surface area, i.e. non-matching land/sea masks, the flux exchange can be non-conservative: When ICON applies air-sea fluxes in ocean areas that are represented by land in GETM, fluxes are not conserved in the atmosphere-ocean-system. Therefore, the differentiation between local and global only makes sense for flux exchange, but not for the interpolation. Throughout the manuscript, we have carefully double checked the correct terminology.

*Furthermore, the authors seem to agree with my analysis when they write "A conservative atmosphere-ocean-system requires the surface area of the sea water fraction in an ICON cell being identical to the corresponding area in the exchange grid, see also next point." and when they discuss the possibility/difficulty to implement mixed land/ocean cells in ICON. Therefore, I strongly suggest that the following sentences be modified so not to mislead the reader:*

- L.4 : *add "locally" before "conservative data exchange via ESMF exchange grids"*

  => For a more clear distinction between interpolation and data exchange, "conservative" has been removed in front of "data exchange". It is now correctly and more clearly written: "ICONGETM is built on the latest NUOPC coupling software with flexible data exchange and conservative interpolation via ESMF exchange grids.

- L.54-55: *at the end of the sentence, add: ", even if the exchange grid cannot force their global conservation if the sea-land masks do not match between the ocean and the atmosphere models.".*

  => We now write: "Moreover, the ESMF exchange grid considers the masking of the original grids, e.g. land/sea masks, and excludes fractions that are not required for the interpolation." We will elaborate on the missing global conservation of fluxes within the developed model system in detail in Sec. 3.4.1, see your next point, and in the discussion.

- L.220: *At the end of this paragraph: "It is also obvious that in case 2, a part of the flux calculated by the atmospheric cells will be lost as it cannot be attributed to any ocean cell in GETM; the global conservation of the fluxes cannot be ensured if the sea-land masks do not match between the ocean and the atmosphere models."*

  => The text has been adapted: "As sketched in Fig. 6, the interpolation of the mean sea level pressure (MSLP) from ICON to GETM is straight-forward in principle, because ICON provides all quantities over the whole domain. However, in case sea surface fluxes are exchanged, there are two issues if the land/sea masks do not match between ICON and GETM. First, there is a physical inconsistency, when surface fluxes parametrized over land cells in ICON are transferred to ocean cells in GETM (case 3). Second, when ICON applies sea surface fluxes in ocean areas that are represented by land in GETM (case 2), the fluxes are not conserved in the global atmosphere-ocean-system. This latter case demonstrates that the conservative interpolation via the exchange grid is not sufficient to guarantee a conservative flux exchange.

- L.351-352: *Modify the sentence for "The applied ESMF exchange grid guarantees a conservative flux exchange, except in the case of non-matching sea-land masks between the ocean and the atmosphere."*

  => Modified.

- L.355-356: *Modify the sentence for "Their calculation directly on the ESMF exchange, even if it cannot solve the problem of different land-sea masks (Balaji et al., 2006) ensures physical consistency ..."*

  => Please see our reply to your next remark.

- L.365-366: *Modify the sentence "Their calculation directly on the ESMF exchange grid also solves the problem of different land/sea masks (Balaji et al., 2006) and ensures physical consistency in the sense that no fluxes calculated over land, i.e. not influenced by the sea surface temperature, are provided to the ocean." by something like: "Even if the ESMF exchange grid does not solve the problem of different land/sea masks (Balaji et al., 2006), it ensures physical consistency in the sense that no fluxes calculated over land, i.e. not influenced by the sea surface temperature, are provided to the ocean. Regarding the problem of matching land/sea masks between the atmosphere and the ocean, it is worth mentioning here that the only way to have a well-posed coupled problem, is to adopt the following best practice, which is applicable only if the atmosphere model can consider water and land sub surfaces. The original sea-land mask of the ocean model should be taken as is. For the atmosphere model, the fraction of water in each cell should be defined by the conservative remapping of the ocean mask on the atmospheric grid. Then, the atmospheric coupling mask should be adapted associating a valid/active index to cells containing at least a fraction of sea. This method ensures that the total sea and land surfaces are the same in the ocean and atmosphere models, allowing global conservation of sea or land integrated quantities. ICON mask was not defined following this best practice (and it would involve some non-trivial*

*modifications to do so, so the global conservation of fluxes cannot be fully ensured in the current coupled model."*

=> We agree with the reviewer that in the present implementation the flux exchange between the atmosphere and ocean is not fully conservative, which is now stated more clearly. We also agree with the outlined approach with mixed land/ocean cells to obtained identical sea surface areas and improved its description. However, following [2] and [1], we also extended and clarified our original argumentation for a conservative flux exchange via the ESMF exchange grid. The paragraphs now read:

"Ideally all fluxes, air-sea and land fluxes, should be calculated directly on one unique ESMF exchange grid in the mediator and applied as boundary conditions to the corresponding individual models [2]. On the exchange grid, a unique land/sea mask of the coupled system can be defined. If the land/sea mask of the exchange grid is adjusted to the mask of the ocean model [1], the associated sea surface areas will be identical. In this case, the conservative interpolation between the exchange and model grids finally offers a fully conservative flux exchange between the atmosphere and ocean, despite originally non-matching land/sea masks in the individual models. Moreover, physical consistency will be ensured in the sense that only air-sea fluxes, i.e. fluxes influenced by the sea surface temperature, are provided to the ocean. If the sea surface area on the exchange grid does not cover the one of the ocean model, creep, nearest neighbor or other extrapolation methods are required to avoid the application of land fluxes to the ocean [see e.g. 4, 3, 5]. In any case, fluxes provided by the mediator can be applied in the atmosphere and ocean over the same period until new fluxes are calculated in the next coupling time step. The flux calculation on the ESMF exchange grid in a central mediator component also offers the most straight-forward extension of the coupled system by models for e.g. waves and sea ice. One drawback of the flux calculation outside the individual models can be stability issues for explicit time stepping schemes or complex coupling implementations for implicit time stepping schemes.

Alternatively, a conservative exchange of air-sea fluxes calculated in the atmosphere model is possible, if the mask of the exchange grid can be emulated in the atmosphere model due to mixed land/ocean cells. For this, the water fraction in each cell must be obtained by conservative interpolation of the sea surface area from the ocean model via the ESMF exchange grid.

In its present state ICON does neither support the described ideal modular coupling nor the alternative. Both approaches require non-trivial modifications to the ICON code. It is expected that they will become available in future releases of ICON, such that the full potential of ICONGETM for a flexible conservative flux exchange via the ESMF exchange grid can be exploited."

Other important comments:

- p.14, l.272: *I don't understand what "a good concurrent load-balancing with minimum idle/waiting times for the single model components was empirically ..." means. Do you mean that the elapsed time for running*

*ICON as single model on 864 processes was almost the same as the elapsed time for running GETM as single model on 384 processes, and therefore you suppose that using these number of processes for each component in the coupled system will lead to minimum idle/waiting time? If so, it should be rephrased for something like "For the present set-up, ICON was run on 864 processes and GETM on 384 processes. It is supposed that this distribution leads to minimum idle/waiting time of any of the component as the elapsed time for running ICON as a single model on 864 processes was about the same than the elapsed time for running GETM as a single model on 384 processes."*

=> No, the load balancing was not predicted based on separate single model runs. From coupled model runs with different processor numbers, we analysed empirically the waiting times of both model components based on the log-files written by ESMF on a higher verbosity level. We now write "For the present setup, a good concurrent load-balancing with minimum idle/waiting times for ICON and GETM was empirically obtained through the log-file time information resulting in 864 and 384 processes, respectively."

- p.15, l.292-293: *The statistics presented are extremely difficult to understand. I suppose that 1.6K/1.5K are for the two-way coupled simulation and that 1.9K/2.0K are for the uncoupled simulation. But e.g. for the two-way coupled, I don't understand what the two numbers (1.6 K and 1.5K) relate to; are these for different averaging periods (maybe 01-10 July and 10 July onward?)? What does "01/10 July 2012 onward" stand for? I have the same remark for the Pearson coefficient. This remark about the need to better quantify the improvement brought by the two-way coupling was done by myself and by the other reviewer. I consider that the answer brought by the author is not satisfying, at least under the current form.*

=> This part of the text has been rephrased: "The average deviation between the modelled and measured temperature in the period from 01 July till 21 July 2012 is decreased from $1.9\,K$ for the uncoupled to $1.6\,K$ for the two-way coupled simulation. This represents an improvement of about 15%. On the other hand, the Pearson correlation coefficient is only slightly improved from 0.7 for the uncoupled to 0.72 for the two-way coupled simulation. Fig. 9 indicates that the coupled ICONGETM system needs some spin-up time to adapt to the coupling, before the improvement with respect to the uncoupled simulation becomes visible. Within the period from 10 July till 21 July 2012, the average deviation between the modelled and measured temperature decreases from $2.0\,K$ for the uncoupled to $1.5\,K$ for the two-way coupled simulation. Thus, after the spin-up, the model results are significantly improved due to the coupling by 25%. The removal of the spin-up period also increases the correlation coefficients to 0.73 for the uncoupled and to 0.75 for the two-way coupled simulation."

Minor comments:

- p.2, l.26: *I suppose that the sentence "The atmosphere model WRF ...with MCT." describes COAWST? If so, it would be clearer by linking the two*

*sentences with something like: "is COAWST (Warner et al., 2010) into which the atmosphere model WRF ... with MCT."*

=> The second sentence has been modified to link the individual models in a more clear way to COAWST.

- p.2, l.27: *"the ocean model" is missing before "ROMS"*

  => Added.

- p.2, l.37: *consider changing "... are the coordinated execution of and the data exchange between the individual models." with "... are the coordinated execution of the individual component models and the data exchange between these models."*

  => The sentence has been rephrased to "... are the coordinated execution of the individual model components and the data exchange among them.".

- p.2, l.39: *It looks like you are doing a distinction between "coupling libraries" and "coupling frameworks" which is fine to me. But ESMF is mentioned as an example of both categories. In the coupling library list, you should replace OASIS by OASIS3-MCT (which has been introduced just above), you should remove "ESMF" and maybe replace it with "YAC (Yet Another Coupler, Hanke et al., 2016)" and put the reference to YAC there in the text (and remove it at l.97).*

  => In order to avoid misleading interpretations, "coupling libraries" is replaced by "coupling software" and "OASIS" by "OASIS3-MCT" as suggested.

- p.2, l.47: *consider changing "from the models are received during runtime" for "are received during runtime from the models"*

  => Changed.

- p.2, l.52: *I don't think that Balaji restricted his definition of an exchange grid to two rectangular grids. Therefore, please consider changing the two sentences for "They have been introduced in Balaji et al. (2006). ESMF implements this functionality to unstructured grids, ..."*

  => Changed.

- p.3, l.66, and p.22, l.372: *why do you call ICON a "next-generation" atmosphere; ICON exists today so it is not a next-generation" model; please consider changing "next-generation" for "state-of-the-art" or something similar.*

  => Modified.

- p.3, l.85: *I think a verb is missing in "can be configured to various models", maybe "can be configured to produce various models"*

  => It is now written: "The atmospheric component of ICON allows various user-configurations for different modelling scenarios, e.g. LES, NWP or climate simulations, by coupling a common dynamical core with different physics packages."

- p.4, l.100: *the link under www.getm.eu does not work (at least for me). Should it be "https://getm.eu" ?*

  => Thanks a lot for this hint. On our side, the link with the http protocol works fine. Maybe your browser only accepts the https protocol. We now changed all links to https and hope that they work for you as well.

- p.6, Table 1 captions: *consider adding "although exchange of state variables is not activated in the simulations reported in this paper" after "same model environment."*

  => Added a similar sentence at the end of the caption.

- p.7 l.133: *consider adding ", although exchange of state variables is not activated in the simulations reported in this paper" after "the exchange of flux data". Make a new sentence for "See Tab. 1 for a list ..."*

  => Modified and added.

- p.7, l.139: *replace "will be used" by "are used".*

  => Changed.

- p.7, l.142: *add "the" before "Initialization phase".*

  => Added.

- p.7, l.159: *why do you write "compare with Fig.1" and not simply "see Fig.1"?*

  => Changed.

- p.8, l.182: *replace "domain distributing" by "domain distribution"*

  => Changed.

- p.13, l.234: *add a comma after "Baltic Sea setup"*

  => Added.

- p.14, l.272: *add a comma after "For the present setup"*

  => Added.

- p.20, l.346: *add "see" before "Fig. 15 D"*

  => Added.

- p.20, l.347: *add "see" before "Fig. 15 A and C"*

  => Added.

**Bibliography**

[Balaji et al.(2006)Balaji, Anderson, Held, Winton, Durachta, Malyshev, and Stouffer] Balaji, V., Anderson, J., Held, I., Winton, M., Durachta, J., Malyshev, S., and Stouffer, R. J.: The Exchange Grid: A mechanism for data exchange between Earth System components on independent grids, in: Parallel Computational Fluid Dynamics 2005, pp. 179–186, Elsevier, doi:10.1016/B978-044452206-1/50021-5, URL `https://linkinghub.elsevier.com/retrieve/pii/B9780444522061500215`, 2006.

[Best et al.(2004)Best, Beljaars, Polcher, and Viterbo] Best, M. J., Beljaars, A., Polcher, J., and Viterbo, P.: A Proposed Structure for Coupling Tiled Surfaces with the Planetary Boundary Layer, Journal of Hydrometeorology, 5, 1271–1278, doi:10.1175/JHM-382.1, URL `http://journals.ametsoc.org/doi/10.1175/JHM-382.1`, 2004.

[Chen et al.(2010)Chen, Campbell, Jin, Gaberšek, Hodur, and Martin] Chen, S., Campbell, T. J., Jin, H., Gaberšek, S., Hodur, R. M., and Martin, P.: Effect of Two-Way Air–Sea Coupling in High and Low Wind Speed Regimes, Monthly Weather Review, 138, 3579–3602, doi:10.1175/2009MWR3119.1, URL `http://journals.ametsoc.org/doi/10.1175/2009MWR3119.1`, 2010.

[Kara et al.(2007)Kara, Wallcraft, and Hurlburt] Kara, A. B., Wallcraft, A. J., and Hurlburt, H. E.: A Correction for Land Contamination of Atmospheric Variables near Land–Sea Boundaries*, Journal of Physical Oceanography, 37, 803–818, doi:10.1175/JPO2984.1, URL `http://journals.ametsoc.org/doi/10.1175/JPO2984.1`, 2007.

[Turuncoglu(2019)] Turuncoglu, U. U.: Toward modular in situ visualization in Earth system models: the regional modeling system RegESM 1.1, Geoscientific Model Development, 12, 233–259, doi:10.5194/gmd-12-233-2019, URL `https://gmd.copernicus.org/articles/12/233/2019/`, 2019.

**Bibliography**

[1]  V. Balaji et al. "The Exchange Grid: A mechanism for data exchange between Earth System components on independent grids". In: *Parallel Computational Fluid Dynamics 2005*. Elsevier, 2006, pp. 179–186. DOI: 10.1016/B978-044452206-1/50021-5.

[2]  Martin J. Best et al. "A Proposed Structure for Coupling Tiled Surfaces with the Planetary Boundary Layer". In: *Journal of Hydrometeorology* 5.6 (2004), pp. 1271–1278. ISSN: 1525-7541. DOI: 10.1175/JHM-382.1.

[3]  Sue Chen et al. "Effect of Two-Way Air–Sea Coupling in High and Low Wind Speed Regimes". In: *Monthly Weather Review* 138.9 (2010), pp. 3579–3602. ISSN: 1520-0493. DOI: 10.1175/2009MWR3119.1.

[4]  A. Birol Kara, Alan J. Wallcraft, and Harley E. Hurlburt. "A Correction for Land Contamination of Atmospheric Variables near Land–Sea Boundaries*". In: *Journal of Physical Oceanography* 37.4 (2007), pp. 803–818. ISSN: 1520-0485. DOI: 10.1175/JPO2984.1.

[5]  Ufuk Utku Turuncoglu. "Toward modular in situ visualization in Earth system models: the regional modeling system RegESM 1.1". In: *Geoscientific Model Development* 12.1 (2019), pp. 233–259. ISSN: 1991-9603. DOI: 10.5194/gmd-12-233-2019.